**Morphological and optical properties of carbonaceous aerosol particles from ship emissions and biomass burning during a summer cruise**
**measurement in the South China Sea**
**Cuizhi Sun[1], Yongyun Zhang[1], Baoling Liang[1,&], Min Gao[1], Xi Sun[1,#], Fei Li[1,4], Xue Ni[1], Qibin Sun[1], Hengjia Ou[1], Dexian Chen[1], Shengzhen Zhou[1,2,3*],**
**and Jun Zhao[1,2,3*]**

[1] School of Atmospheric Sciences, Guangdong Province Key Laboratory for Climate Change and Natural Disaster Studies, and Southern Marine Science and

Engineering Guangdong Laboratory (Zhuhai), Sun Yat-sen University, Zhuhai, Guangdong 519082, China

[2] Guangdong Provincial Observation and Research Station for Climate Environment and Air Quality Change in the Pearl River Estuary, Zhuhai, Guangdong

519082, China

[3] Key Laboratory of Tropical Atmosphere-Ocean System, Ministry of Education, Zhuhai, Guangdong 519082, China

[4] Xiamen Key Laboratory of Straits Meteorology, Xiamen Meteorological Bureau, Xiamen, Fujian 361012, China

[&] Now at Guangzhou Environmental Monitoring Center, Guangzhou, Guangdong 510060, China

[#] Now at Centre for Isotope Research (CIO), Energy and Sustainability Research Institute Groningen (ESRIG), University of Groningen, Groningen 9747

AG, the Netherlands
*Correspondence to*: Jun Zhao (zhaojun23@mail.sysu.edu.cn) and Shengzhen Zhou (zhoushzh3@mail.sysu.edu.cn)
**Abstract.** Carbonaceous aerosols constitute a crucial component of atmospheric marine aerosols among which black carbon (BC) and brown carbon (BrC)
are important contributors to light absorption and hence the positive climatic radiative forcing in the marine atmosphere. We conducted a month-long (May
05–June 09, 2021) onboard sample collections and online measurements of carbonaceous aerosols to characterize their morphological and optical properties
during a ship cruise in the South China Sea (SCS), covering a marine region of 11.9–24.5 °N and 111.1–118.2 °E. Single particles were collected by a single
particle sampler and offline analyses were performed using a transmission electron microscope (TEM) coupled with energy dispersive X-ray spectroscopy
(EDS). Online measurements of BC in $PM_{2.5}$ were made by a seven-wavelength aethalometer and organic carbon (OC)/elemental carbon (EC) mass
concentrations were measured by a semi-online OC/EC analyzer. Feret diameters of the single particles during navigation and stop showed size distributions
with the lognormal fitting peaks at 307 and 325 nm, respectively. The fresh (without coating) and aged BC particles (after removal of coating by the electron
beams in TEM) showed same median fractal dimensions (1.61), in contrast to their different median lacunarities (0.53 vs 0.59). The aged BC particles showed
narrower Feret diameters (229–2557 nm) during navigation than those (78-2926 nm) of freshly-emitted BC from the own ship during stop. Moreover, tar
balls, as one important component of single particles from ship emissions and as the tracer of biomass burning, were identified with geometrical diameters of
160–420 nm in the TEM images. The EDS analyses showed those tar balls are mainly mixed with sea salt, organics, BC, and sulfate. We also found a
significant fraction of aged BC in various mixing states (core-shell, embedded) with other components of the aerosol particles after long-range transport.
The campaign was further divided into several periods (before monsoon period, BMP; transition monsoon period, TMP; after monsoon period, AMP; and
ship pollution period, SPP) according to the wind direction during monsoon and the own ship pollution. The median absorption Angström exponent (AAE)
values derived from all wavelengths were 1.14, 1.02, 1.08, and 1.06 for BMP, TMP, AMP and SPP, respectively. Particularly, a median AAE value of 1.93
was obtained during two significant biomass burning events. These results showed that biomass burning (BB) and fossil fuel (FF) combustion contributed to
18–22% and 78–82% of all the BC light absorption without the two intense biomass burning events, during which  BB and FF accounted for 42% and 58%,
respectively. The two BB events originated from the Philippines and Southeast Asia before and after the summer monsoon. Our results demonstrated that BC
can serve as the core of aged particles but the fractal dimensions of BC aggerates were subject to little variation; moreover, such BC particles become much
more aggerated after aging in the marine atmosphere, which further affects the light absorption of the BC particles in the SCS.

## 1 Introduction

Carbonaceous aerosols (e.g., organic carbon (OC), elemental carbon (EC)/black carbon (BC)) profoundly impact regional and global climate (Corbin et al., 2019; Lu et al., 2020; Rabha and Saikia, 2020). As an important component of carbonaceous aerosols, BC can serve as a tracer of anthropogenic pollution once emitted from the incomplete combustion of fossil fuels and biomass burning. Moreover, BC particles are generally soot-aggerated with graphene-like layer microstructures which can be observed under electron microscopy (Adachi et al., 2019). BC and EC are two components of carbonaceous aerosols that are measured differently. BC is typically quantified based on its light-absorbing properties, while EC is measured using thermal-optical methods (Duarte et al., 2021). However, EC can also be referred to as graphitic carbon or soot, with some overlap in their definitions. Another important component of carbonaceous aerosols, BrC, represents a series of light-absorbing organic compounds, contributing significantly to the light absorption of atmospheric aerosols (Wang et al., 2020b). BrC and BC show different light absorption patterns as a wavelength function. BrC and BC can be distinguished by measuring the absorption spectra of aerosol particles at different wavelengths (Andreae and Gelencsér, 2006; Bond et al., 2013; Laskin et al., 2015; Li et al., 2020; Yus-Díez et al., 2021). Tar balls are commonly used as typical tracers of biomass and biofuel burning due to their composition of amorphous carbon. These particles also belong to BrC because they are light absorbing organics (Adachi et al., 2019; Hand et al., 2005). Spherical tar balls emitted from biomass burning have been observed in cases of both wild fire burning (Adachi et al., 2019) and laboratory generated tar ball particles (Tóth et al., 2014). In the atmosphere, biomass burning produces a significant amount of tar balls, which are not deliquescent but can absorb water at high relative humidity (RH = ~80%), thereby affecting their ability to scatter and absorb light (Hand et al., 2005).

The optical properties of BC and BrC particles are affected by several factors including the emission source, coating component, particle size, morphology, and mixing state of the particles (Wei et al., 2020). The BC configuration in the single particles would influence their radiative effects (Luo et al., 2021). For example, core-shell BC particles show enhanced light absorption compared to bare BC particles, especially when BC particles are coated with absorptive materials such as BrC (Budhavant et al., 2020; Cappa et al., 2012; Shamjad et al., 2012; You et al., 2016). "Lensing effect" refers to the absorption enhancement if BC is coated with non-absorbing organic or inorganic materials (Luo et al., 2021; Yang et al., 2009). In contrast, if the BC coating materials are highly absorptive, no absorption enhancement may occur at shorter visible and UV wavelengths, a phenomenon known as "shielding effect". The shielding and lensing effects depend on the coating thickness over BC (Lack and Cappa, 2010). When BC is well internally mixed with BrC, its total absorption enhancement becomes smaller than the enhancements of not well mixed counterparts due to the absorptive coating that acts as a shield (Feng et al., 2021). Moreover, it is impossible for BC and other materials to be homogeneously distributed.

The extent to which BC and BrC contribute to light absorption in atmospheric aerosols can be assessed using the absorption Ångstrom exponent (AAE)
(Wang et al., 2020a). AAE is a parameter used to quantify the spectral dependence of aerosol light absorption. It is calculated by fitting a power-law relationship
between the aerosol absorption and wavelengths over a given spectral range. The AAE is used to identify sources and types of aerosols and a higher AAE
value is associated with sources such as biomass burning or urban pollution, while a lower AAE value suggests absorption by larger particles, such as mineral
dust or sea salt (Blanco-Donado, 2022; Duarte et al., 2021). However, many factors such as mixing state, coating, particle size, refractive index, wavelength,
and emission source, would affect the AAE values for BC and BrC aerosols, leading to large variations among different studies (Moschos et al., 2021). For
example, a previous study showed that the AAE values derived from wavelengths of 405 and 781 nm are very sensitive to refractive index and particle
diameter (Chylek et al., 2019). The AAE values of 0.8−1.6 at 470 and 950 nm are attributed to traffic emissions and fuel combustion (Ezani et al., 2021).
Comparatively, those AAE values can be as large as 2.0 for ship emissions (Helin et al., 2021). Moreover, the recommended AAE value for fossil fuel (FF)
and biomass burning (BB) is 1 and 2 (or higher), respectively (Liu et al., 2023). Other AAE values were also found in previous studies for FF (0.9) and BB
(1.68) (Zotter et al., 2017) or FF (1.2) and BB (2.2) as the mostly used optical pair (Milinković et al., 2021). The AAE values of 1.4 and 1.7 for BC and BrC
were set to be the lower and upper limits in the modelling study of biomass burning particles mixed with BC and BrC (Chylek et al., 2019). However, the use
of constant AAE values for calculating the BC fractions from BB and FF led to large uncertainties without knowledge of the core size or coating thickness of
the BC particles (Virkkula, 2021). Currently, the effect of the light absorption is not well known for the carbonaceous particles in the marine atmosphere due
to scarce ship-based measurements.
The optical properties of BC and BrC particles can also be investigated through fractal dimension ($D_f$) analysis based on the fractal properties of BC
aggerates. $D_f$ illustrates how particles aggregate and grow and it can be determined through boxing counting calculation, ensemble method, or soot parameter
method with TEM images (Pang et al., 2022). The $D_f$ values are mainly related to emission sources and aging process of the particles. Previous studies showed
that the $D_f$ values of fresh BC particles tend to be small but become larger after aging because the particles are more compact due to coatings (Luo et al., 2022;
Wang et al., 2017). $D_f$ values of 1.8 and 2.6 were used respectively for fluffy and compact BC particles in a numerical study to investigate the impact of the
BC morphology on light absorption (Luo et al., 2021). Laboratory experiments simulating wildfires showed that the $D_f$ values of freshly emitted BC were in
a range of 1.74−1.92 (Chakrabarty et al., 2006), compared to the range of 1.67−1.83 from a field study of the Las Conchas fire (China et al., 2013). A similar
range of the $D_f$ values of 1.67−1.93 were found at a remote site in the southeastern Tibetan Plateau (Wang et al., 2017). The obtained $D_f$ values for the traffic
emissions were as large as 3 (Wei et al., 2020). In addition to the emission sources and aging process, $D_f$ is also dependent on the particle size. A previous
experimental study found that for polystyrene latexes (PSL) particles, the $D_f$ values decreased with the increase of particle size up to 200 nm (Wu et al.,
2013b). Nevertheless, knowledge of the fractal dimension for carbonaceous particles in the marine atmosphere is currently very limited, hindering our ability
to understand the aging process and the optical properties of these particles.
In the past years, carbonaceous aerosols in the marine atmosphere have been extensively studied on regional and global scales, focusing on the transport of
anthropogenic emissions to the sea areas. The BC background concentrations in Antarctic and Arctic regions are below 20 ng m$^{-3}$ (Fossum et al., 2022). The
BC outflows from Asia to the Pacific Ocean exhibit seasonal variations and originate from anthropogenic and biomass-burning sources in China, Siberia, and
Southeast Asia (Matsui et al., 2013). Ship-based BC and EC measurements reveal significant influence of continental transport on remote oceanic regions,
including the Bay of Bengal (Kedia et al., 2012), Indian Ocean (Kompalli et al., 2021), Southern Indian Ocean and the Southern Ocean (Ueda et al., 2018),
North Sea (Bencs et al., 2020), Antarctic (Chaubey et al., 2013; Schmale et al., 2019), North Pacific (Taketani et al., 2016; Xing et al., 2014), Arctic (Pankratova
et al., 2021; Sharma et al., 2019), Northeast Atlantic (Fossum et al., 2022), the Yellow Sea (Kwak et al., 2022), and Western Pacific (Ma et al., 2022). However,
to our knowledge, the BC mass concentrations have been found to vary significantly across different oceans and seasons, with levels from 3 to 2800 ng m$^{-3}$
and being influenced by anthropogenic activities and seasonal factors. The online BC measurements in the South China Sea (SCS) region are limited. An
early study reported BC concentrations on Yongxing Island during the rainy season (May 16–June 20, 2008) and the dry season (Dec. 12, 2008–Jan. 8, 2009),
with average concentrations of 0.54 and 0.67 μg m$^{-3}$, respectively (Wu et al., 2013a). Recent studies conducted at coastal sites in the SCS found that BC
concentrations are strongly impacted by land anthropogenic emissions (Wang et al., 2022). The time-resolved BC concentration varies with the vertical heights
(Sun et al., 2020c) and the carbonaceous materials of OC and EC account for 31–62% in PM$_{2.5}$ (Yan et al., 2018). Nevertheless, quantification of the light
absorption potential of BC and BrC aerosols remains challenging due to the limited knowledge regarding the morphology, particle size, and mixing state of
carbonaceous particles in the SCS (Kompalli et al., 2021). Furthermore, the atmosphere in the SCS is typically influenced by the southwesterly monsoon from
May to August (Wang and Wu, 2020), which affects the air masses from Southeast Asia. In this study, we conducted ship-based measurements of BC, OC/EC,
and single particle sampling during summer (May 05–June 09, 2021) in the SCS. The morphology (i.e., the fractal dimension and the size of the single BC
particles) and light absorption properties of carbonaceous particles were characterized. The source origins, relationships between the D$_f$ and BC size, as well
as the impact of summer monsoon on the light absorption of the BC particles are discussed.

**2 Methods**

**2.1 Cruise route and instrumentation**

The cruise measurements were carried out from May 5 to June 9, 2021, covering a marine area of 11.9−24.5 °N and 111.1−118.2 °E in the SCS. Single

particles were collected on the TEM grids (3.05 mm I.D., copper meshed and covered with lacey carbon film) located on the front deck during ship navigation

and stop using a single-stage particle sampler (DKL-2, Genstar Electronic Technology Co., Ltd., China) which is the same as other studies (Chen et al., 2023;

Dong et al., 2018; Liu et al., 2021; Pang et al., 2022). The sampling flow rate and time were set at 1 L min$^{-1}$ and 10 min, respectively, for each collection. The

nozzle diameter of this single-cascade impactor is 0.3 mm. The particles with aerodynamic diameters above 0.2 μm were collected with a collection efficiency

of 50%, assuming a particle density of 1.5 kg m$^{-3}$ (Marple and Olson, 2011). More details can be found in the supplementary information (Section 1 of SI).

The mixing state and morphology of the single particles were analyzed utilizing a transmission electron microscope (TEM, FEI Tecnai G2 Spirit, Holland)

operated at an accelerating voltage of 120 kV, in conjunction with an energy dispersive spectrometer (EDS, Bruker Nano GmbH Berlin, Esprit 1.9, Germany)

for elemental analysis. The thickness of the EDS detector (type XFlash 5060) is 0.45 mm with a Si dead layer of 0.029 mm. Notably, in the EDS spectra,

when analyzing particle composition, Cu should be excluded, and a considerable level of C and Si should be observed in the background signals due to the

presence of Si in the detector, Cu and C in the TEM grid. The substrate holder of TEM was tilted 25° for thorough inspection during imaging and EDS analysis.

The sampling inlets were installed on the bow of the research vessel with a height of ~ 15 m above sea level. The own ship emissions (e.g., engine, cooking,

etc.) were exhausted from the chimney on the stern with a linear distance of ~ 22 m to the inlets. The BC mass concentrations were measured by an aethalometer

(Model AE33, Magee Scientific, USA) with a time resolution of one minute (Drinovec et al., 2015). Note that the BC mass concentrations derived from AE33

are referred to as equivalent BC mass concentrations due to the light absorption of both BC and BrC at 880 nm. The sampling air was regulated by a PM$_{2.5}$

cyclone (BGI Inc., Waltham, MA, USA) and subsequently dried by a Nafion dryer (Model MD-700 series, Perma Pure Inc., USA) with a relative humidity

below 40% through the filter tape (type 8060) at a sample flow rate of 5 L min$^{-1}$. Data corrections were made for the employed Aethalometer AE33, considering

the multiple scattering parameters (C($\lambda$)=1.39) for the used filter type, the leakage factor ($\zeta$=0.01), and the compensation parameters (K$_{min}$=-0.005, K$_{max}$=0.015).

The measured attenuation at seven wavelengths (7 channels) is used to determine the wavelength-dependent absorption coefficient. The mass specific

absorption cross-sections (MAC, $\sigma_{air}$) applied in the BC calculations were 18.47, 14.54, 13.14, 11.58, 10.35, 7.77, and 7.19 m$^2$ g$^{-1}$ for wavelengths of 370,

470, 520, 590, 660, 880, and 950 nm, respectively (Ausmeel et al., 2020). The measured values at 880 nm (channel 6) are for black carbon concentration

calculation, and at 370 nm (channel 1) for UV particulate matter (UVPM) concentration (Drinovec et al., 2015). The detection limit of AE33 aethalometer is

approximately 0.03 µg m$^{-3}$ for 1-min integration period and below 0.005 µg m$^{-3}$ for 1-hour integration period. The instrument was automatically calibrated by
zero air every day. Notably, significant spikes were observed during periods when the ship was stationary, when it was travelling at low speeds, and when the
wind was blowing from the stern of the vessel.
The OC/EC concentrations were measured by a semi-continuous OC/EC analyzer (Model-4, Sunset Laboratory Inc., USA) based on the optical attenuation
and thermo-optical transmittance methods (Geron, 2009) under the NIOSH 5040 thermal-optical protocol (Lappi and Ristimaki, 2017). Similarly, the sampling
air passed through a PM$_{2.5}$ cyclone (BGI Inc., Waltham, MA, USA) and was dried by a Nafion dryer (Model MD-700 series, Perma Pure Inc., USA) with a
relative humidity below 40% at a flow rate of 8 L min$^{-1}$. The air then passed through a denuder for the removal of volatile organic compounds (VOCs) and
the particles were collected on the quartz filter with 45-min accumulation and 15-min analysis. The instrument was calibrated with the standard sucrose
solution as recommended. The manufacturer-claimed detection limits are 0.4 and 0.2 µg m$^{-3}$ for OC and EC, respectively (Brown et al., 2019). However,
several previous studies showed that these values may vary substantially in a range of 0.04−2 and 0.001−0.5 µg m$^{-3}$ for OC and EC, respectively, due to the
artifact of the quartz filters (Bao et al., 2021; Bauer et al., 2009; Chen et al., 2017; Jung et al., 2011; Karanasiou et al., 2020; Park et al., 2018; Zhang et al.,
2021). Here, we estimated the instrument noise (including contamination) of 0.15 and 0.012 µg m$^{-3}$ for OC and EC based on 26 effective blank measurements
with 3 times the standard deviation (3σ) during the campaign. The limit of detection (LOD) for OC and EC is 0.18 and 0.19 µg m$^{-3}$, respectively, calculated
as three times the standard deviation of replicate measurements of a standard sucrose solution with a carbon content of 10.516 µg m$^{-3}$. The Sunset OC/EC
analyzer also measures optical EC based on the transmission of 660 nm wavelength light through the quartz fiber filter employed for sampling, similar to the
AE33 for optical BC measurements. Optical EC is defined as the apparent EC on the filter based on the measured apparent absorbance and the fixed absorption
coefficient according to the user's manual of the Sunset OC/EC. Both our study and a previous study (Brown et al., 2019) showed that the optical EC
concentrations from Sunset were comparable with the BC concentrations from AE33. Note that the resultant optical EC concentrations from the instrument
output may be overestimated due to the limitation of the filter-based optical measurements.
The measurements of solar radiation (SR), temperature (T), pressure (P), relative humidity (RH), relative wind direction (RWD), and relative wind speed
(RWS) were provided by the automatic weather station (AWS430, Vaisala Inc., Finland) (Song et al., 2022) equipped on the front deck of the research vessel.
This station comprises a range of integrated sensors, including a wind speed and direction sensor (model WMT702), a temperature and humidity sensor (model
HMP155), and an atmospheric pressure sensor (model BARO-1). The cruise route for ship navigation is from the global positioning system (GPS) onboard
the ship (Seapath 330+, Kongsberg Inc., Norway).

## 2.2 Data analyses and processing

### 2.2.1 Analyses of single particles

A total of 34 samples (15 during navigation and 19 at stop) were analyzed and more than 20 bright-field TEM images were randomly captured for each sample except for those at the center of the grids where particles were easily overlapped. A total of 15624 single particles were statistically analyzed to obtain morphology information (i.e., the Feret diameter, area, perimeter) for each particle using the software ImageJ (1.53q, National Institute of Health, USA) (Cheng et al., 2021). In the analysis of particle size, the Feret diameter is defined as the distance between the parallel tangential lines that constrain the particle perpendicularly. In this study, we applied the Feret diameter as the longest distance between any two points along the boundary of the selected particles. Moreover, we utilized "geometrical diameter" to describe the size of tar balls with circular shape, which signifies the distance between two points located on the surface of a geometric shape, with this line passing through the shape's center. Using "Geometrical diameter" is suitable to quantify the size of the observed tar balls which excluded any coatings or additional materials. Specifically, we employed TEM data acquisition software to measure the geometrical diameters of observed tar balls. The $D_f$ values of the BC particles were estimated using the boxing counting method using the plugin Fraclac. An example was given in the SI (section 2, Fig. S1) to show how to calculate $D_f$ using the software. A detailed description of the procedure using the boxing counting method and the software ImageJ can be found in the SI. The $D_f$ values are very sensitive to the fill extent and sizes of the particles. A previous study showed low fractal dimensions when the particles contain void volumes (Peyronel et al., 2010).

The own ship emissions can be identified using various measures, for example, high CO, $NO_x$ concentrations (Sun et al., 2020b), high BC concentrations (Alroe et al., 2019; Shank et al., 2012), regular cooking emissions (Cai et al., 2020), and wind speeds/directions between the ship stop and start operation (Ausmeel et al., 2020; Kwak et al., 2022). The contribution of ship emissions to BC sources on the marine atmosphere depends on engine types, operation modes, fuel types, and loadings (Gagne et al., 2021; Jiang et al., 2018; Karjalainen et al., 2022; Lack and Corbett, 2012; Wu et al., 2021; Zhao et al., 2020). Here, we classify two sampling modes (navigation vs stop) of single particle analyses according to ship operation, and relative wind direction/speed. In this study, the relative wind direction/speed is relative to the ship heading. The navigation mode is constrained by the relative wind direction of $0-80°$ or $280-360°$, and the relative wind speed greater than 5 m s$^{-1}$, averaged for every 10 minutes (consistent with the collection time of TEM samples). The stop mode is set with the relative wind direction of greater than 80° and less than 280°, or the relative wind speed lower than 5 m s$^{-1}$. The navigation mode samples are mainly from marine air and air masses of long-range transport while the stop mode collected air masses which are mixed with the own ship emissions. The wind direction (speed) and relative wind direction (speed) are calculated by Eq. (1) (Aijjou et al., 2020).

$$V_R = \sqrt{V_s^2 + V_w^2 + 2 * V_s * V_w * \cos\alpha} \qquad (1)$$

where $V_R$ is the relative wind direction (speed), $V_s$ is the ship direction (speed), $V_w$ is the true wind direction (speed), α is the angle between the ship heading and the true wind direction.

The temporal profiles of ship heading directions, and relative wind direction/speed are shown in the SI (section 3, Fig. S2). Details of the two sampling modes (navigation vs stop) on a vector average of 10 minutes are listed in Table S1 and Fig. S3. Here, we distinguished the own ship emissions (research vessel) from those of other ships or long-range transport based on the following criteria: low relative wind speed (< 5 m s$^{-1}$), relative wind direction encompassing ship exhaust (80–280°), and a substantial AE33-derived hourly averaged BC mass concentration (>2 μg m$^{-3}$). Other ship emissions far from the research vessel are treated as a part of the transported air masses in this study.

**2.2.2 BC, OC, EC and optical EC data**

In this study, BC data obtained from the AE33 instrument are referred to as BC, while data from the OC/EC analyzer is expressed as thermal OC, thermal EC, and optical EC. Here, we averaged the BC mass concentrations over one minute and excluded those below the detection limit of 0.03 μg m$^{-3}$ to minimize the variations. We also removed the own ship emissions which are characterized by spikes in particle number concentrations according to the wind directions (Fossum et al., 2022). BC mass concentration was calculated using Eqs. (2, 3) which are cited from the AE33 aethalometer user's manual (Ver 1.54).

$$ATN = -100 * \ln(I/I_0) \qquad (2)$$

where ATN is optical attenuation, $I_0$ is reference signal, I is spot signal.

$$BC = \frac{S*(\Delta ATN_1/100)}{F_1(1-\zeta)*\sigma_{air}*C*(1-k*ANT_1)*\Delta t} \qquad (3)$$

where BC is black carbon concentration, S is spot area, $F_1$ is measured flow, ζ is leakage factor, $\sigma_{air}$ is the mass absorption cross-section (MAC), C is multiple scattering parameter, k is compensation parameter, and t is time.

AAE was calculated according to Eq. (4) using the light absorption at wavelengths of 470 and 950 nm, which are built-in algorithms in the AE33 aethalometer as described elsewhere (Helin et al., 2021; Kang et al., 2022; Milinković et al., 2021; Zotter et al., 2017). This method serves as a two-composition source apportionment for BC emitted from fossil fuels and biomass burning (AAE model), which applied AAE=1 for fossil fuel and AAE=2 for biomass. The calculations for BC(BB) and BC(FF) are shown in Eqs. (5–7) which are referred to the AE33 aethalometer user's manual and publication (Sandradewi et al., 2008). The optical absorption coefficient is the sum of biomass burning and fossil fuel burning contributions. Basic equations are using Beer-Lambert's Law.

$$AAE = -\frac{\ln\frac{\sigma_{abs}(\lambda_1)}{\sigma_{abs}(\lambda_2)}}{\ln\frac{\lambda_1}{\lambda_2}} \qquad (4)$$
where $\sigma_{abs}$ is aerosol absorption coefficient, $\sigma_{air}$ is mass absorption cross-section (MAC), $\sigma_{abs} = BC*\sigma_{air}$. $\lambda_1 = 470$ nm and $\lambda_2 = 950$ nm.
$$\frac{\sigma_{abs}(470\,nm)_{FF}}{\sigma_{abs}(950\,nm)_{FF}} = \left(\frac{470}{950}\right)^{-AAE_{FF}} \qquad (5)$$
$$\frac{\sigma_{abs}(470\,nm)_{BB}}{\sigma_{abs}(950\,nm)_{BB}} = \left(\frac{470}{950}\right)^{-AAE_{BB}} \qquad (6)$$
$$\sigma_{abs}(\lambda) = \sigma_{abs}(\lambda)_{FF} + \sigma_{abs}(\lambda)_{BB} \qquad (7)$$
where $\sigma_{abs}(470\,nm)_{FF}$ and $\sigma_{abs}(950\,nm)_{FF}$ are the aerosol absorption coefficients at wavelengths of 470 and 950 nm for fossil fuel (FF), $\sigma_{abs}(470\,nm)_{BB}$ and
$\sigma_{abs}(950\,nm)_{BB}$ are the aerosol absorption coefficients at wavelengths of 470 and 950 nm for biomass burning (BB), $AAE_{FF}$ and $AAE_{BB}$ are equals to 1 and 2,
respectively.
Alternatively, AAE can be obtained from the negative slope of linear regression between the log-transformed $\sigma_{abs}$ and all the wavelength spectra so that
hourly AAE values (all $\lambda$) can be obtained following a similar method in Retama et al. (2022). Details are shown in the SI (Section 4, Figure S4). Here, we
define Delta-C as the difference between the concentration derived from the aforementioned UVPM data (at 370 nm) and BC concentration (at 880 nm). This
Delta-C parameter was employed as an indicator of smoke from biomass burning in previous wood biomass burning studies (Harrison, 2020; Zhang et al.,

214    2017).

The OC and EC (thermal) concentrations lower than the instrument noise (0.15 and 0.012 $\mu$g m$^{-3}$ for OC and EC, respectively) were excluded. Additional
data were removed for those with laser correction factors below 0.88 and calibration peak areas lower than the initial calibration levels (within 10%), and a
total of 551 h data were used for further analysis. In comparison, the Sunset optical EC (at 660 nm) is generally consistent with the Magee AE33 aethalometer
derived BC (at 880 nm) within 9% (Brown et al., 2019) which is shown in Section 3.3. The EC concentration data from Sunset were considered as ship
pollution and were discarded when the BC concentrations from the AE33 aethalometer were higher than 2 $\mu$g m$^{-3}$, in addition to those with relative wind
directions of 80–280° regardless of the BC concentrations as mentioned before.
**2.2.3 HYSPLIT backward trajectory and MODIS fire data**
The backward trajectories were calculated using NOAA HYSPLIT (Hybrid Single-Particle Lagrangian Integrated Trajectory) (Version 5) at heights of 100,
500 and 1000 m above sea level (AGL). Daily meteorological data with 1.0°×1.0° spatial resolution for trajectory calculation were downloaded from the
global data assimilation system (GADS) (ftp://arlftp.arlhq.noaa.gov/pub/archives/gdas1/). Here, we calculated the 72-h back trajectories of air masses arriving
at the single particle sampling sites along the cruise route.
Moderate Resolution Imaging Spectrometer (MODIS) data are available from the Near real-time MODIS Collection 6 products
(https://firms.modaps.eosdis.nasa.gov/download/). Here, we selected a region of 102–127 °E and 0–30 °N fully covering the campaign area. The number of
fire hotspots was counted each day during the campaign with a confidence level of higher or equal to 80% as recommended (Giglio et al., 2020). A detailed
description of the fire detection algorithms is available online (https://earthdata.nasa.gov/what-is-new-collection-6-modis-active-fire-data).
**3 Results and discussion**
**3.1 Overview**
Figure 1 shows the time series of ship cruise route and the single particle sampling locations during ship navigation (marked as solid triangles) and stop
(marked as open squares) over the SCS during the campaign. The cruise sequences are AB→C→D→EB→D→A, with AB and EB being non-stop cruise,
otherwise the ship stopped occasionally along the arrow routes for other research tasks. Figure 2 shows the time series of the meteorological variables (i.e.,
solar radiation (SR), temperature, pressure, relative humidity (RH), wind direction (WD), and wind speed (WS) during the whole campaign (May 05–June
09, 2021). The measurements were conducted mostly on sunny days prior to June 02 as shown by the SR data. Subsequently, it became rainy and cloudy due
to the summer monsoon in the SCS. One notable meteorological feature during the campaign was the occurrence of the summer monsoon starting from May
27 close to the site **E**, during which (May 27–June 01), an increase of RH (~9% from campaign-averaged 78.7% to monsoon period-averaged 85.6%) and a
slight decrease of pressure (~0.2% from 1007.4 to 1005.4 hPa) were observed. Meanwhile, the wind directions were mainly southerly during this period and
later changed to southwest.
It should be noted that Typhoon 202103 (CHOI-WAN) travelled across B→D, resulting in a bulge in the middle of the cruise route to avoid the typhoon
during June 03–05, 2021. The typhoon track is available online with the last accessed date Mar. 25, 2023: http://agora.ex.nii.ac.jp/digital-
typhoon/summary/wnp/s/202103.html.en. The typhoon was initiated at 02:00 local time on May 31 and dissipated at 14:00 on June 05, 2021 (Figure S5). It
passed over our cruise route from June 03 to June 05, 2021. While no significant increase of absolute wind speed was seen in Figure 2, a significant increase
of relative wind speed was shown in Figure S2, along with an obvious decrease of atmospheric pressure during the typhoon period (Figure S5). The measured
relative humidity increased from May 27 to June 01, prior to the presence of the typhoon, which can be attributed to the decrease of ambient temperature
during this period.
Figure 3 shows the time series of the mass concentrations of carbonaceous aerosol components in $PM_{2.5}$ (i.e., BC, UVPM, OC, and EC) during the whole
campaign. Frequently high spikes of the mass concentrations of carbonaceous aerosol components were observed due to the ship pollution from the research
vessel. We notice that ship pollution was significantly pronounced on the first two days after the ship left the harbor and on the last 3–4 days before the ship
returned to the harbor, during which the spikes of BC and UVPM concentrations were measured by the Magee AE33 aethalometer with the relative wind
direction of 80–280°. Before May 08 and after June 05, higher UVPM, OC, and EC concentrations were observed, which can be attributed to significant fresh
ship emissions from the research vessel, as evidenced by simultaneous higher BC concentrations. Similar spikes in BC concentrations were observed during
other measurement periods, either preceding or following the monsoon period, which were caused by emissions from the frequent stops and starts of the ship.
Note that no significant diurnal trend for OC was observed during those aforementioned periods.
Figure 4 shows the time series of fire spots distribution and the 72-h backward trajectories at three AGLs (100, 500 and 1000 m) over the SCS during the
campaign. Only several backward trajectories are shown to avoid massive overlapping. Several fire spots located in the sea were attributed to oil or natural
gas drilling processes which generate thermal energy, combustion, and exhaust. Such processes included the prevalent hydrocarbon exploration and production
activities in this region. A comprehensive cartographic representation of these endeavors within the SCS can be accessed via the online platforms
(https://amti.csis.org/south-china-sea-energy-exploration-and-development/). Note that since the ship moved along the cruise route, the air mass backward
trajectories also changed with the movement of the ship. For example, significant biomass burning was detected in Laos, northern Vietnam, and the Philippines
during May 15–24, as indicated by the corresponding fire spots. However, the back trajectories to the sampling route (C→D) during this period were mainly
from the Philippines.
Here, we classified the campaign period into several groups based on the cruise route, change of wind direction during monsoon, backward trajectories,
and ship pollution, as listed in Table 1: (1) BMP-1 (before monsoon period 1), AB route mainly with northeast wind direction during May 05–09; (2) BMP-
2, B→C→D route close to the Philippines primarily with southeast wind direction during May 10–22; (3) BMP-3, D→E close to mainland China with the
same wind direction as BMP-2 during May 23–26; (4) TMP (transition monsoon period), EB route with south wind direction during May 27–Jun 01; (5) AMP
(after monsoon period), B→D→A route with southwest wind direction during June 02–09; (6) SPP (ship pollution period), ~35% of the online measurement
data could be attributed to this category in this study due to the interference from the research vessel own emissions.
**3.2 Single particle analysis of BC and tar balls**
Particle size distribution, composition, and size-dependent BC fractals were investigated based on TEM images. The Feret diameter is commonly used in

272 microscopy for particle size analysis (Zefirov et al., 2018). The size distribution of all the single particles from the analyzed TEM images is depicted in Fig.

273 5. The distribution is represented with histograms starting at 50 nm, a width interval of 20 nm, and a bin number of 200. The choice of bin width may vary

274 depending on cases but it is close to the quotient value of the square root of the measured particle number divided by the overall width of the distribution

275 (Pabst and Gregorova, 2007). Moreover, lognormal fitting is used for the peak size identification of particle size distribution (Rice et al., 2013). Figure 5a

276 shows a fitted peak Feret diameter of 307 nm for a total of 6613 particles from 15 samples during navigation, while a fitted peak Feret diameter of 325 nm

277 was obtained for a total of 9011 particles from 19 samples during stop (Fig. 5b). Note that we could not successfully obtain a bimodal or multi-peak fit for the

278 data of the stop cases using multi-peak fitting function in the Igor Pro software, as shown in Figure S6. Hence, we believe that single peak fitting best described

279 the distribution in our stop cases, as illustrated in Figure 5. Particles collected during navigation were predominantly aged at high wind speeds, while particles

280 during stop were mixed significantly with freshly emitted particles from the own ship and from other merchant ships or those from long-range transport,

281 possibly leading to the variation of the size distribution. Although the bimodal distribution was observed from particles in the indoor air, which was likely

282 caused by fresh emissions and secondary formation (Pipal et al., 2021), we did not obtain significant bimodal peaks for both navigation and stop particles.

283  We obtained characteristic values for the particle shape descriptors such as circularity ($0.7 \pm 0.2$) and aspect ratio ($1.2 \pm 0.3$) for all the particles collected

284 during navigation and stop, implying that these particles are not perfectly spherical and may vary in their mixing states. Figure 6 (top images) shows a

285 comparison of the mixing states during navigation (a-c) and stop (d-f) from typical BC TEM images. The BC particles collected during navigation are in the

286 embedded (a), external (b), or core-shell (c) mixing states classified with the methods which are based on single particle analysis of island and mountain

287 samples across East China Sea and Japan (Adachi et al., 2014; Sun et al., 2020a). More TEM images for the heavily coated internal and external BC particles

288 from navigation can be found in the SI (section 7, Fig. S7). The EDS analysis showed that the single particles during navigation were predominantly composed

289 of carbon (C), oxygen (O), sulfur (S), potassium (K), sodium (Na), chloride (Cl), magnesium (Mg), and calcium (Ca) (Fig. S8), indicating that those BC

290 particles were coated with sulfate, sea salt, and organics. Furthermore, small externally mixed BC particles can be transported over the sea and easily coated

291 during long-range transport. Under the TEM electron beam, these coated volatile components were easily vaporized to expose the BC fractal frame (Fig. S7d-

292 f).

293  Comparatively, a mixture of aged BC particles and much larger fresh BC particles as well as smaller scattered BC particles during stop were found (Fig.

294 6d-f), which were likely emitted from other ships (Fig. 6d) and the research vessel (e, f). These TEM images showed that the compressed BC particles are

295 typically more aged and atmospherically processed, while the fractal BC particles are fresh. Moreover, EDS analysis showed that sulfate formed from aqueous

296 processes and less viscous organic coating indicate an aging process. Those BC particles with Feret diameters larger than 2 µm during stop were fractal

aggerates which could unlikely survive due to deposition during long-range transport. In addition, heavily coated internal BC particles were found during stop
due to the mixing between ship pollution and the marine air (Fig. S9). Moreover, such particles could also be condensation of organics during the cooling
process after they were emitted from the ship engine. The bottom panels of Fig. 6 (a-f) show the corresponding images obtained by boxing counting in fractal
analysis with the resultant $D_f$, Feret diameter (D), Lacunarity (L), and sampling number underneath for each TEM sample image. Figure 7 shows $D_f$ and L as
a function of D for some representative BC particles during both navigation and stop. The BC particles showed narrower Feret diameters (229–2557 nm)
during navigation than those (78-2926 nm) of BC from the own ship during stop. The $D_f$ values during navigation were in a range of 1.28–1.77 with a median
of 1.61, while the $D_f$ values during stop were 1.43–1.76 with a median of 1.61, indicating no significant differences of $D_f$ for the exposed BC particles during
navigation and stop. Note that the particles in Figure 7 include pure BC and BC without thick coatings. These particles were exposed to the electron beam and
volatile coatings were removed so that the morphology of BC was clearly shown regardless of the mixing state of the original BC particles (Figure S7). Most
BC particles were below 1 μm in Feret diameter during navigation (Figure 7), while their sizes cover a wide range below 3 μm during stop, implying that the
aged BC particles become smaller after long-range transport. Despite only a total of 134 BC data points shown in Figure 7, the results are still statistically
meaningful due to the wide range of BC sizes covered in our analysis. Note that the size change of a BC particle cannot be determined because the original
size of the particle is unknown before the removal of the coatings. Comparatively, the lacunarities during navigation (0.34–0.82, median: 0.53) and stop (0.34–
0.92, median: 0.59) were slightly different, with the former being smaller than the latter, indicating that the lacunarity tended to become smaller (~10%) after
coating or aging of the BC particles.
Tar balls were frequently observed during the campaign with an estimated sample fraction of about 11.8%. Fractal-like tar ball aggregates were usually
found in wildfire smokes (Girotto et al., 2018); however, in this study, spherical tar ball particles were observed in the marine atmosphere and were mixed
with sea salt (Fig. 8a and d for TEM image and EDS spectrum, respectively), organic carbon and sulfate (Fig. 8b and e) from the samples collected on May
27 during navigation. In contrast, the particles collected on June 01 were found to be amorphous carbon agglomerates (Fig. 8c and f) which were referred to
OC. During these days, the wind directions were from the southwest, with air masses originating from both the Philippines and Southeast Asia. The shape
difference between the tar ball spheres and the amorphous carbon agglomerates may be related to the type of biomass burning or the origin of the ship engines.
Similar particle morphologies were found in other studies on brown carbon during aircraft measurements over the Yellow Sea in 2001 (Zhu et al., 2013). Tar
balls mixed with BC during stop were also observed (Fig. S10), with geometrical diameters of 160–420 nm, much larger than nano-soot spheres (40–50 nm)
(Fig. S11). In comparison, the laboratory-generated tar balls were measured to have AAE values of 2.7–3.4, with an average of 2.9 at 467–652 nm (Hoffer et
al., 2016).

### 3.3 Light absorption of carbonaceous aerosols

The BC concentrations measured by the Magee AE33 aethalometer agree excellently with the optical EC concentrations obtained from the Sunset OC/EC analyzer, as evidenced by a linear regression coefficient of 0.97. The BC measurements obtained from the AE33 instrument do not agree with the OC, EC values, yet their overall trends exhibit consistency. However, the BC concentrations were considerably higher than the thermal EC concentrations, exhibiting linear regression coefficients of 1.66 and 1.55, respectively. These findings, presented in Fig. S12 of Section 8 in the SI, are in line with previous research conducted by Brown et al. (2019). The OC/EC ratios can be used as an indicator for the source origins of the air masses. Figure 9 shows the distribution of the OC/EC ratios and the corresponding EC concentrations. The median OC/EC ratios are 8.14, 5.20, 6.35 and 2.63 for the classified periods BMP, TMP, AMP, and SPP, respectively. Notably, EC median mass concentrations (0.24, 0.25 and 0.17 $\mu g\ m^{-3}$) for the marine air masses during BMP (0.013−0.69 $\mu g\ m^{-3}$), TMP (0.015−0.60 $\mu g\ m^{-3}$), AMP (0.014−0.74 $\mu g\ m^{-3}$) were lower than the median concentration (1.70 $\mu g\ m^{-3}$) during SPP. Compared with Figure 9d, the scattered higher OC/EC ratios in Figure 9a/b/c are caused by the very low EC concentrations. The presence of extremely low EC concentrations, often falling below or near the detection limit, can introduce discrepancies in the calculation of the OC/EC split, ultimately resulting in inaccurate EC concentrations (Bauer et al., 2009). In addition, this study revealed a significant variation in EC concentrations during SPP, ranging from 0.15 to 22.8 $\mu g\ m^{-3}$. Previous studies showed that OC/EC ratio could be characterized by various sources, ranging from 1.37–1.71 for residential cookstoves, 1.63–2.23 for rural emissions, 1.05–1.24 for diesel exhaust, and 0.80–1.12 for urban environments (Khan et al., 2012). A low OC/EC ratio (<3) corresponded to the predominant contribution of the primary OC in submicron particles reported in a previous study in the Southern Indian Ocean, Northern Indian Ocean and Bay of Bengal (Neusüß, 2002). Here, the median OC/EC ratio of 2.63 during SPP is much higher than the characteristic values of diesel combustion, most likely because the sample air during SPP is composed of marine air and the own ship exhaust. Our results are consistent with a recent study which showed that the diesel combustion from ships accounted for 15% of BrC in the total light absorption at a coastal site in Shanghai during June–July, 2021 (Kang et al., 2022). In contrast, the OC/EC ratios during other periods (i.e., BMP, TMP and AMP) were even much higher (5.20–8.13), indicating that the aerosols were highly aged during the long-range transport of biomass burning aerosols. This is also consistent with our recent study in the SCS which showed that during monsoon periods in the summer of 2019. The biomass burning organic aerosols became aged through atmospheric processes during transport (Sun et al., 2023).

The long-range biomass burning transport affects the air mass in the SCS. Figure 10 illustrates the wavelength-dependent mass concentration measured by the AE33 aethalometer during the campaign, showing (a) an example of a ship plume, and (b, c) two significant biomass burning events during BMP (BB-1: 6:00−7:00 on May 15 and 15:00−22:00 on May 16) and during TMP (BB-2: 15:00 on May 30−00:00 on May 31). The ship plumes, predominantly emitted

from fossil fuel combustion, showed similar absorption at all seven wavelengths. In contrast, significant absorption at low wavelengths was detected during
the biomass burning events, a phenomenon also observed in other field measurements in urban cities and towns where air masses were susceptible to biomass
burning (Zhang et al., 2017). A comparison of the two methods for AAE calculation is presented in the SI (Section 4, Fig. S4). The fitting results demonstrate
that the AAE calculated for all wavelengths was lower than the AAE calculated for only 470 and 950 nm wavelengths. The fitting slope is 0.78, and the
determination coefficient ($R^2$) is 0.98, indicating a strong correlation between the two methods.
Figure 11 shows the hourly averaged AAE derived from all wavelengths as a function of the BC concentrations by AE33 aethalometer with the median
(range) AAE values of 1.14 (0.57−1.48), 1.02 (0.51−1.36), 1.08 (0.54−1.42), and 1.06 (0.65−1.37), respectively, for the classified periods (BMP, TMP, AMP,
and SPP), and the corresponding BC median (range) mass concentrations of 0.28 (0.033−1.17), 0.14 (0.042−2.86), 0.17 (0.055−1.08), and 3.01 (0.21−36.5)
$\mu g \, m^{-3}$, respectively. Like EC, ship pollution led to emissions of high BC concentrations, reaching as high as 36.5 $\mu g \, m^{-3}$. The median BC concentrations
decreased significantly during TMP and AMP, likely due to the increase of the RH which further increased the scavenging of the BC particles during navigation
as reported previously (Girach et al., 2014). Note that the biomass burning events were excluded from the above calculations and are further discussed below.
During the biomass burning events, the correlations of AAE with AE33 BC and Delta-C concentrations are respectively shown in Figs. 11 and 12,
characterized by very high median AAE values (1.85 and 1.86, respectively for BB-1 and BB-2) and BC concentrations (1.93 and 1.67 $\mu g \, m^{-3}$). The BC mass
concentration ranged from 1.45 to 3.62 $\mu g \, m^{-3}$ during biomass burning events based on light absorption at wavelength of 880 nm. The mass concentration in
Figure 10 corresponds to BC mass concentration obtained at each wavelength. We have emphasized that BC mass concentration in this study is equivalent
BC at individual wavelength. Notably, efficient light absorption of BrC in the range at 370–660 nm was observed during the biomass burning events, while
no significant wavelength-dependent BC concentrations were found during the own ship pollution (Fig. 10a). The AAE values below 1 in Figure 11 are not
noises, in some cases due to aerosols from fossil fuel (Ezani et al., 2021) and in other cases, they can be even lower than 0.5 when paired with wavelengths
of 470 and 660 nm (Laing et al., 2020). The higher AAE values imply much stronger absorption of non-BC light-absorbing particles (BrC) at shorter (UV-
vis) wavelengths, which mainly originated from the biomass burning emissions (Ponczek et al., 2022). Moreover, the median OC/EC ratios were 5.03 and
5.29 respectively for the two biomass burning events, even much higher than those for SPP (Fig. 9). The 72-h backward trajectories also showed that the BB-
1 air masses mainly originated from the Philippines while the BB-2 air masses were from the mainland Vietnam, both with high densities of fire spots (Fig.
4). The AAE values were also highly correlated with the Delta-C values with a determination coefficient ($R^2$) of 0.92 (Fig. 12), further demonstrating a
significant contribution of BrC to the AAE enhancement. In addition, we further correlated the observed high AAE values with the Delta-C values for the two
biomass burning events and confirmed that these high AAE values (1.45–3.62) were indeed attributed to biomass burning rather than ship emitted tar balls

371 which covered an AAE range of 2.7–3.7 at 405 and 781 nm wavelengths in a previous wood burning study (Chylek et al., 2019).

372 Our study found that the AAE values from all wavelengths for the marine atmosphere and ship pollution were 1.02–1.14 and 1.06, respectively, except for

373 a higher AAE value (1.93) during the two biomass burning events. The AAE values for ship pollution are dependent on the fuel types and loading conditions

374 (Laskin et al., 2015). For example, heavy fuel oil operated at high loads can result in AAE values (at 470/950 nm, and hereafter unless specified) of 2.0, while

375 the intermediate fuel oil has an AAE value of 1.3 at high loads (Helin et al., 2021). In addition, the presence of tar balls may contribute to the enhancement of

376 BrC absorption as mentioned in Section 3.2, as tar balls from ship emission have higher AAE values (2.5–6 depending on the wavelengths) (Corbin et al.,

377 2019). The occurrence of tar balls in this study was about 12% in the analyzed single samples. These tar balls were likely aged during long-range transport

378 from biomass burning and hence affected the light absorption of BrC in the SCS.

379 **3.4 BC sources from fossil fuel vs biomass burning**

380 The source origins of BC particles can be investigated using the AAE model. In the model, we employed respectively the characteristic AAE values of 1 and

381 2 for FF and BB to calculate their corresponding BC concentrations, namely BC(FF) and BC(BB). Figure 13 shows the time series of BC(FF) and BC(BB)

382 for different classified periods. The BC(FF) and BC(BB) values were much higher before the monsoon than during/after the monsoon, except for the periods

383 during BB-1 and BB-2 with significantly high BC(FF) and BC(BB) values (peaks > 1 $\mu$g m$^{-3}$), while high BC(FF) values were seen during SPP. Table 2

384 summarizes the average concentrations and the ranges of BC(FF) and BC(BB), along with their corresponding fractions. In general, both the average BC(FF)

385 and BC(BB) values were low during BMP, TMP, and AMP, compared to those during the biomass burning events and SPP. BC(FF) contributed over 80% of

386 ship pollution during SPP, whilst the BC(BB) could contribute to more than 40% of the total black carbon during the two BB events. We hence conclude that

387 fossil fuel combustion is the major contributor to the light absorption of BC except during the seasonal biomass burning events and biomass burning can have

388 a profound contribution to the BC light absorption in the SCS.

389 Active biomass burning pollution during January–May in Southeast Asia occurs routinely because of crop residue and sugar cane burning. A previous study

390 showed that during dry and wet seasons, the annual contribution of BC(BB) was 11% and 30% respectively in the Peninsular India (Soyam, 2021) based on

391 the two-component AAE model (Drinovec et al., 2015; Yus-Díez et al., 2021). Table 3 summarizes the BC concentrations, AAE values, and the corresponding

392 fraction of biomass burning (or fossil fuel) in previous and present studies for the marine atmosphere conducted at coastal sites or ship-based cruise

393 measurements using the AE series instrument. The BC(FF) and BC(BB) fractions of 58% and 42% were obtained respectively during the two BB events,

394 while they accounted for 78–83% and 18–22% during other periods, similar to those found at the coastal site in Central Adriatic (79% and 21%), and

significantly different from those reported at a coastal site in the East China Sea (Yu et al., 2018). However, observation data are still lacking on the contribution
of BC from fossil fuel vs biomass burning in the sea regions which warrants more future studies during different seasons.
**3.5 Limitation of this study**
This cruise campaign for carbonaceous aerosols has several limitations which might need to be aware of due to the time and area coverage constraints. The
presence of other light-absorption aerosol components, polluted dusts, oil and gas drilling emissions, as well as fishery policy may contribute to the
uncertainties in the AAE model used for the BC source apportionment in this study. Firstly, the composition of aerosols and refractive index may strongly
affect the AAE calculation. The source apportionment of BC for biomass burning and foil fuel is based on the AAE two-component model which only
considered BC(BB) and BC(FF) as the light absorption materials. An AAE range of 0.9–1.4 is used for pure BC from foil fuel emissions, while it is 1.68–2.2
for biomass burning as mentioned earlier. The current AAE two-component model does not include other potential light-absorbing materials, such as mineral
dust and biological particles (Pileci et al., 2021). Interestingly, two types of possible biological particles were observed during the campaign (Fig. S13, in the
SI, section 9). A similar type of biological particles was observed and identified as brocosomes in another campaign near the East China Sea (Fu et al., 2012).
However, more future studies are needed to identify the types and species of biological particles and to evaluate their contributions to the light-absorption.
Secondly, based on the Cloud-Aerosol Lidar & Infrared Satellite Observation (CALIPSO) data on May 15 and June 07 when the orbit just passed over the
SCS region and the Southeast Asia (Fig. S14 and S15, in the SI, section 10), we found the presence of polluted dust in the vertical profile over the Philippines,
Indonesia, Thailand and Malaysia. Long-range transport of dust may affect our measured AAE data. Thirdly, oil and natural gas drilling (Liu and Li, 2021) is
active in the SCS region and the distribution map is available online (https://www.oilmap.xyz/). These activities potentially contribute to the BC emissions
(Cordes et al., 2016), and these BC are similar to those of continental emitted BC from incomplete burning of oil or natural gas.
Lastly, Chinese fishery policy enacts fishing prohibition for about three and a half months every year in the SCS during summer which corresponds to
May 01–Aug 16 in the year of 2021 in 12°N within our campaign region. Therefore, the cruise measurements mainly captured ship emissions from the
commercial ships in the SCS whose routes are available online (https://www.marinetraffic.com/en/ais/home/centerx:116.6/centery:20.5/zoom:4). The average
BC mass concentrations (~0.2 $\mu$g m$^{-3}$ for BC(FF) and 0.05–0.08 $\mu$g m$^{-3}$for BC(BB) are only limited to data of about a month and the coverage area. Hence,
more future measurements covering more seasons and wider areas are needed to better understand the morphology and optical properties of the carbonaceous
aerosols in the SCS.

## 5. Conclusions

As important components of carbonaceous aerosols, BC and BrC in the marine atmosphere may exert significantly positive climatic radiative forcing through light absorption on the regional and global scales. However, quantification of their absorption potential is tremendously challenging due to little knowledge on the microphysical properties, such as morphology, particle size, and mixing state of BC or BrC in the marine region such as SCS. This ship-based study is intended to investigate the morphological and optical properties of the BC particles in the SCS during summer using a combined online aethalometer, semi-online OC/EC analyzer, and offline TEM/EDS analyses. The results showed that the lognormal fitted Feret diameter distribution of the single particles peaks at 325 nm when the ship stopped, while it peaks at 307 nm when the ship navigated. This minor difference in the size distribution could be attributed to the distinguishable air mass origins of the own ship emissions for the former and the mixed other ship emissions and long-range transport for the latter. Furthermore, the Feret diameters of the single particles spread much more narrowly during navigation (229-2557 nm) than those of the freshly emitted particles during stop (78-2926 nm). In addition, the two types of single particles haves same median fractal dimension values (1.61) but different lacunarity values (0.53 vs 0.59), indicating their different aging degrees. The aged BC particles are present in various mixing states (core-shell, embedded, external) with other aerosol components after long-range transport. Interestingly, a fraction of single particles were identified as tar balls with geometrical diameters of 160–420 nm which were primarily mixed with sea salt, organics, BC, and sulfate, and those tar balls were found to originate from either ship emissions or long-range transport of biomass burning.

Since the marine atmosphere is mainly subject to the influence of biomass burning and fossil fuel combustion, a two-component (BB and FF) AAE model was employed to evaluate the source contributions to the light absorption of the BC particles. The modelling results indicated that BB and FF contributed respectively to 18–22% and 78–82% of all the BC light absorption except for a substantial percentage of 42% for BB (hence 58% for FF) during the two observed significant biomass burning events. The results from trajectory calculations showed that biomass burning was predominantly from the Philippines and South East Asia before and after the summer monsoon during the cruise campaign. However, this highly simplified two-component AAE model may have substantial uncertainties in the evaluation of the source contributions when other sources of BC particles were present and those included dust, biological materials, oil and gas drilling emissions during the measurements. Nevertheless, this study demonstrates that emissions from commercial ships and biomass burning from Southeast Asia contribute to the enhanced light absorption of the BC particles in the SCS, especially during the crop harvest seasons before monsoon, and the aged BC particles became more aggerated after long-range transport of air masses containing biomass burning emissions.

## Author contributions

JZ, CZS, and SZZ planned the cruise campaign. CZS, YYZ, BLL, MG, and XS performed the measurements. CZS performed the data analysis and wrote the original draft. CZS and DXC performed funding acquisition. JZ and SZZ performed funding acquisition and supervision. All authors reviewed and edited the manuscript.

## Declaration of competing interest

The authors declare no conflict of interest.

## Data availability

Data for figures and tables, along with raw data from online measurements and offline analyses of this study are available from JZ via zhaojun23@mail.sysu.edu.cn upon request. The supplementary data are available online at xxx.

## Acknowledgements

We acknowledge supports from the Guangdong Basic and Applied Basic Research Foundation (Grant No. 2022A1515011864; 2021A1515011556), the National Natural Science Foundation of China (NSFC) (Grant No. 42175115; 42205108), and Guangdong Major Project of Basic and Applied Basic Research (Grant No. 2020B0301030004). the Science and Technology Program of Guangdong Province (Science and Technology Innovation Platform Category, no. 2019B121201002), Guangdong Province Key Laboratory for Climate Change and Natural Disaster Studies (Grant 2020B1212060025). This study was also supported by the Southern Marine Science and Engineering Guangdong Laboratory (Zhuhai) through its South China Sea Monsoon Experiment Cruise (No. SML2021SI1002). Additional support from the crew of the vessel "Tan Kah Kee" is greatly acknowledged.

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

**Table 1.** Classification of the campaign period during May 05–June 09, 2021.

| Name | Date | Cruise route | Monsoon | Wind direction |
|------|------|-------------|---------|----------------|
| BMP-1 | May 05–09 | AB | before | 0–90°, northeast |
| BMP-2 | May 10–22 | B→C→D | before | 90–180°, southeast |
| BMP-3 | May 23–26 | D→E | before | around 180°, southeast |
| TMP | May 27–Jun 01 | EB | transition | around 180°, south |
| AMP | June 02–09 | B→D→A | after | 180–270°, southwest |
| SPP* | Screened | Screened | N/A | Screened |

* Ship pollution period is screened based on BC concentration and relative wind direction as mentioned in the method section.

**Table 2.** Source apportionment of BC based on the two-component AAE model.

| Period | a.BC(FF)[*] (μg m$^{-3}$) | a.BC(BB)[*] (μg m$^{-3}$) | r.BC(BB)[*] (μg m$^{-3}$) | r.BC(FF)[*] (μg m$^{-3}$) | f.BC(FF)[*] (%) | f.BC(BB)[*] (%) |
|---|---|---|---|---|---|---|
| BMP | 0.2 ± 0.1 | 0.08 ± 0.06 | 0–0.9 | 0–0.3 | 77.9 ± 5.8 | 22.1 ± 5.8 |
| TMP | 0.2 ± 0.3 | 0.06 ± 0.1 | 0.02–1.8 | 0–1.1 | 82.2 ± 6.2 | 17.8 ± 6.2 |
| AMP | 0.2 ± 0.1 | 0.05 ± 0.05 | 0.01–0.9 | 0–0.3 | 80.8 ± 4.0 | 19.2 ± 4.0 |
| SPP | 4.4 ± 5.7 | 0.7 ± 0.9 | 0.2–32.8 | 0.04–10.3 | 83.0 ± 6.7 | 17.0 ± 6.7 |
| Bio. [**] | 0.8 ± 0.3 | 0.8 ± 0.4 | 0.05–1.2 | 0–1.5 | 58.1 ± 16.7 | 41.9 ± 16.7 |

[*] a represents average, r for range, f for fraction;
[**] Bio. stands for the two biomass burning events as noted in the main text.

**Table 3.** Summary of AAE values, mass concentrations, and the relevant fractions for the BC particles in $PM_{2.5}$ from coast site and cruise measurements using
the AE series aethalometer.

| Region | Time | AAE values (at wavelengths, nm) | BC avg. conc. ($\mu g\ m^{-3}$) | Fraction (%) | Reference |
|---|---|---|---|---|---|
| East China Sea[c] | May, 2017 | 0.9–1.3 (370–950) | 0.8–3.6 | 2.5–11 or 45–60 for BrC | (Yu et al., 2018) |
| Central Adriatic[c] | Feb.–Jul., 2019 | 1.25–1.49 (470/950) | 0.57 ± 0.64 | 79 for BC(FF) 21 for BC(BB) | (Milinković et al., 2021) |
| Bay of Bengal[c, n] | Dec., 2008–Jan., 2009 | 1.81–1.98 (370–950) | 5.1 ± 3.0[c]; 2.5 ± 1.4[n] | <10 for BC(FF) >85 for BC(BB) | (Kedia et al., 2012) |
| South China Sea[n] | Sep.–Oct. 2019 | - | 1.9 ± 0.4 | - | (Wang et al., 2022) |
| South China Sea[c] | Dec. 2017 | 1.2–1.5 (375/880) | 6.6–4.9 | - | (Sun et al., 2020c) |
| South China Sea[n] | May–Jun. 2021 | 1.02–1.14 or 1.93 (370–950) | 0.33 ± 0.38 | 78–82 for BC(FF) 18–22 for BC(BB) BB events: 58 for BC(FF) 42 for BC(BB) | This study |

[c] coastal site measurement;
[n] cruise (remote) measurement.

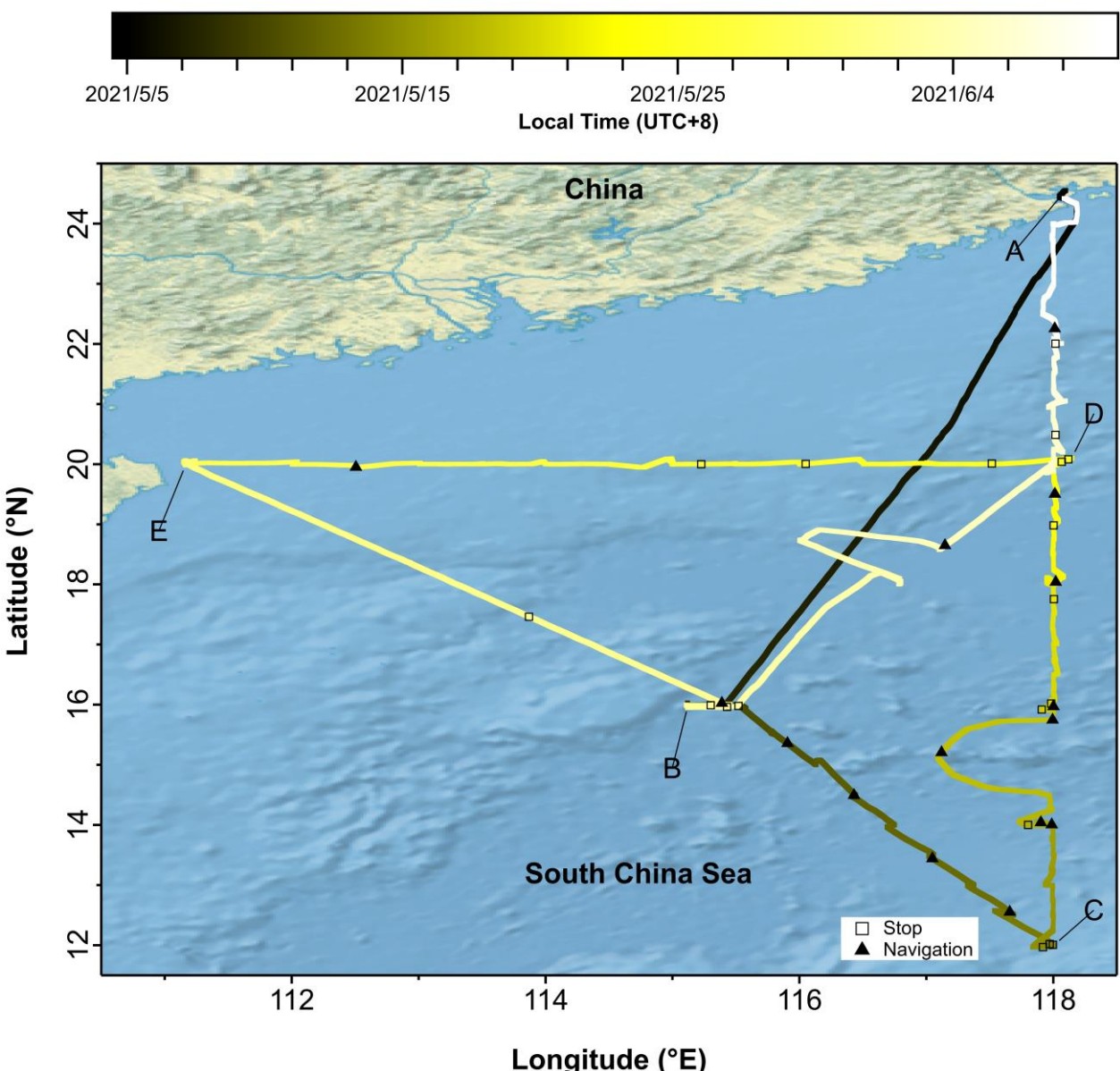


**Figure 1. Map of the cruise route for the campaign in the South China Sea during May 05–Jun 9, 2021. The ship route is dated by the intensity bar at the top. The open**
**squares and solid triangles indicate the single particle sampling location, collected during stop and navigation, respectively.**

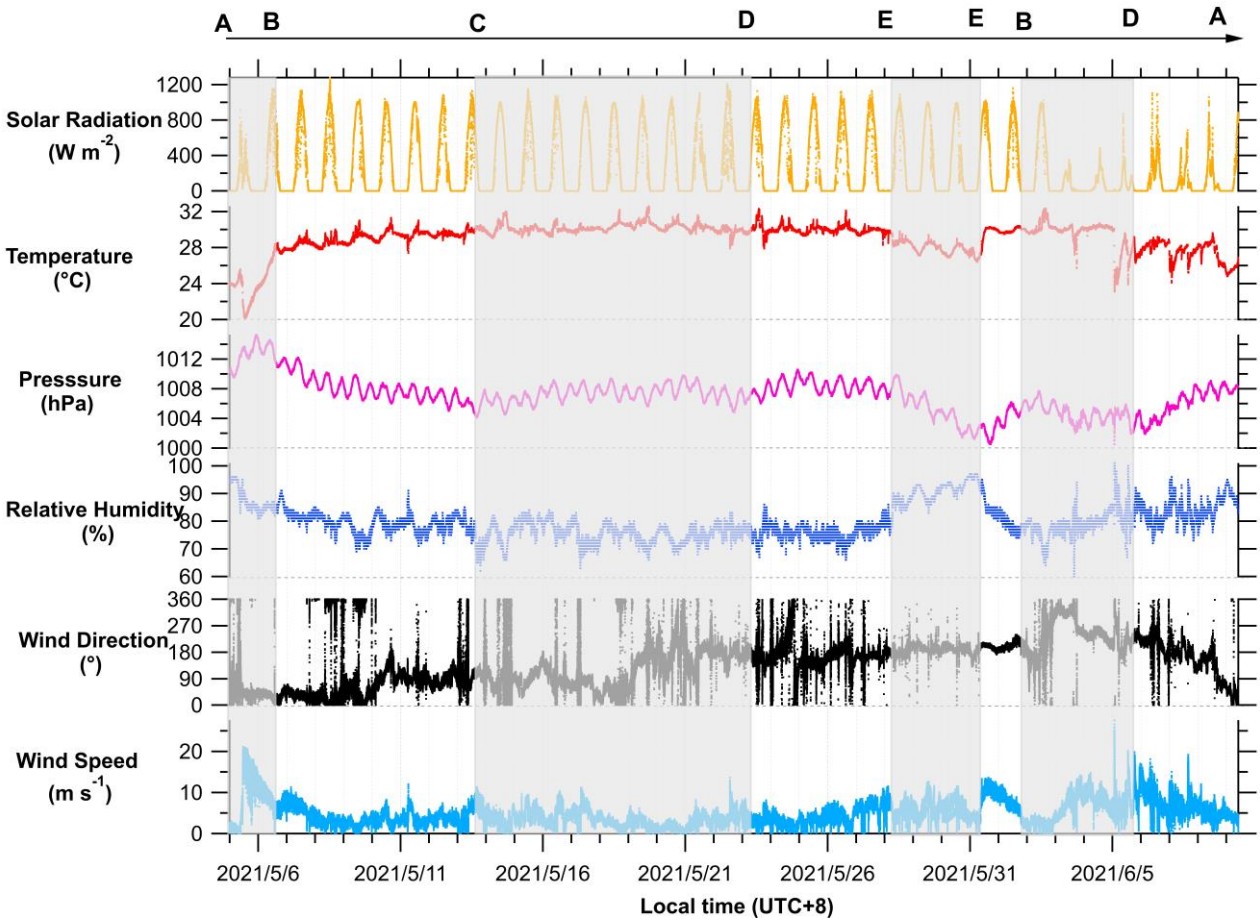


**Figure 2.** Time series of meteorological variables of solar radiation (SR), temperature, pressure, relative humidity (RH), wind direction (WD), and wind speed (WS)

during the campaign. The time resolution is 1 min for all the data except for WS and WD (3 sec). All data points are shown in dots style. The shaded and unshaded areas
sequentially indicate the cruise routes from AB, B to C, C to D, D to E, E to E (ship stop), E to B, B to D, and D to A, as marked in the Figure 1.

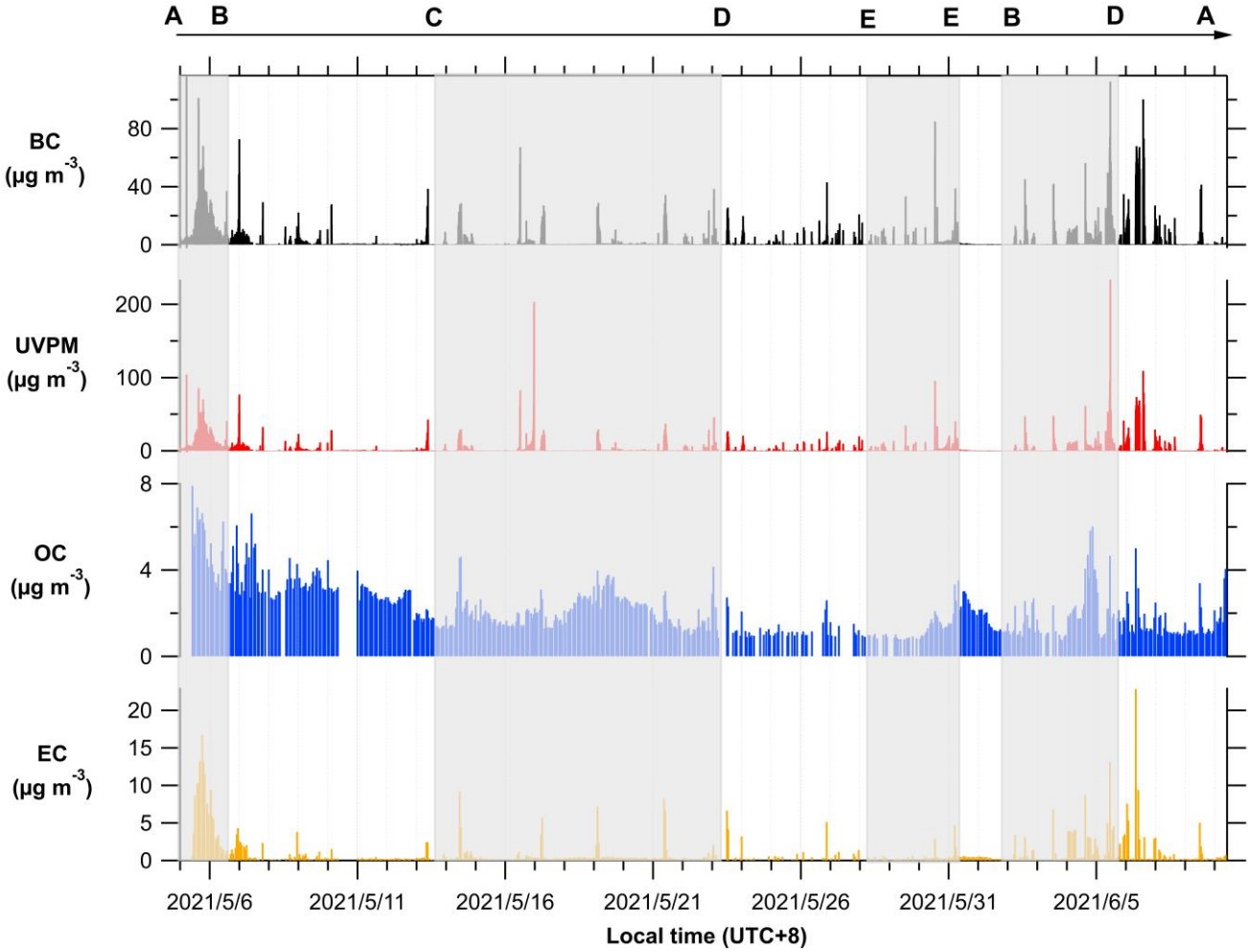


**Figure 3.** Time series of the mass concentration of black carbon (BC), ultraviolet particle matter (UVPM), organic carbon (OC) and elemental carbon (EC) in PM$_{2.5}$

during the campaign. The time resolution is 1 min for all the data except for OC and EC (1 h). All data points are shown in stick-to-zero style. The shaded and unshaded

areas sequentially indicate the cruise routes from AB, B to C, C to D, D to E, E to E (ship stop), E to B, B to D, and D to A, as marked in the Figure 1.

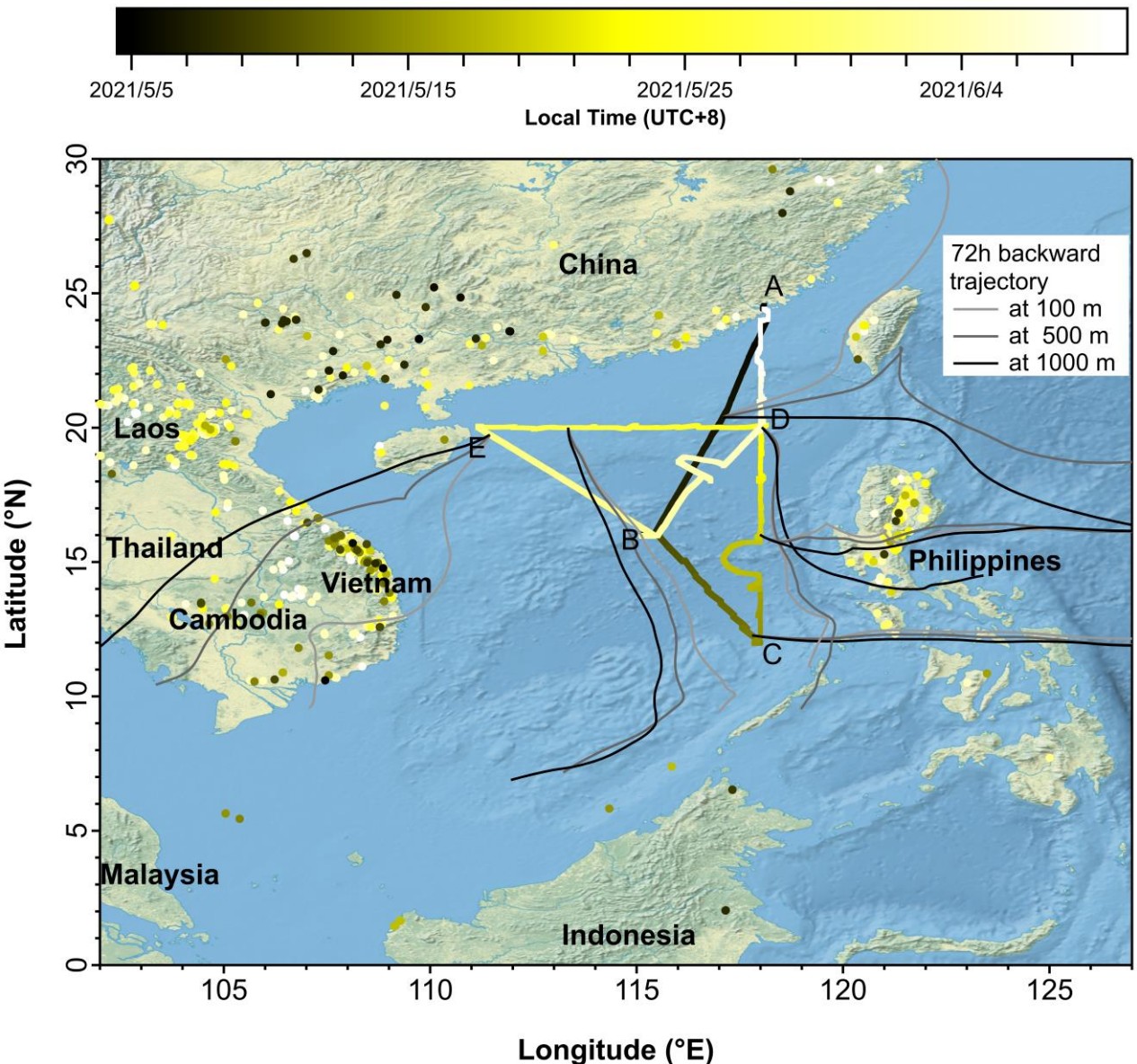

**Figure 4.** The time series of fire spots distribution and the 72-h backward trajectories over the South China Sea (SCS) during the campaign. The solid circles were dated by the intensity bar at the top, representing the detected fire spots using MODIS satellite with a confidence threshold of >80%. The light grey, grey and black 72-h backward trajectories were obtained at AGL heights of 100, 500 and 1000 m using the HYSPLIT model.

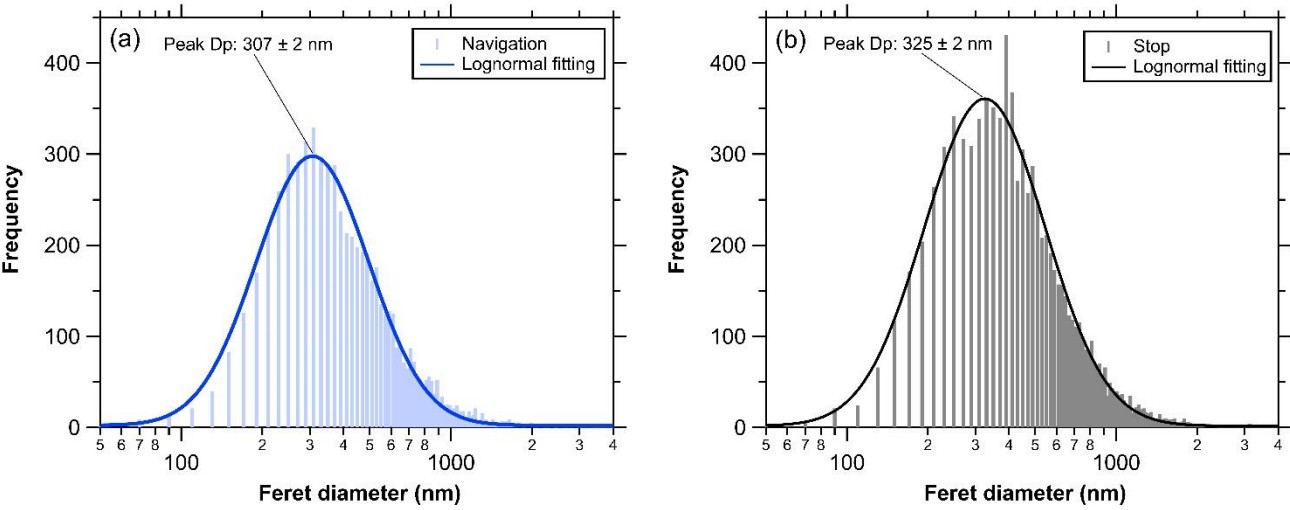


**Figure 5.** Lognormal fitting of particle size distribution using Feret diameter determined from the TEM images with the ImageJ analysis during navigation (a) and stop

(b). The histograms are set with a bin starting at 50 nm, a bin width of 20 nm, and a total bin number of 200.



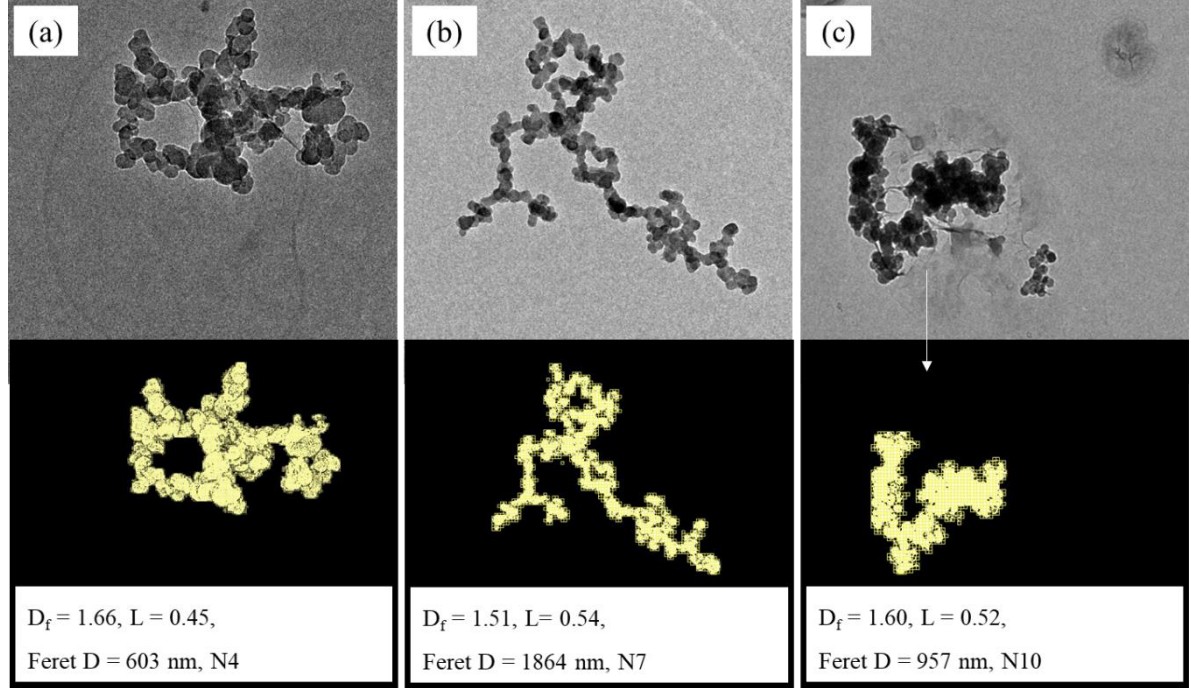


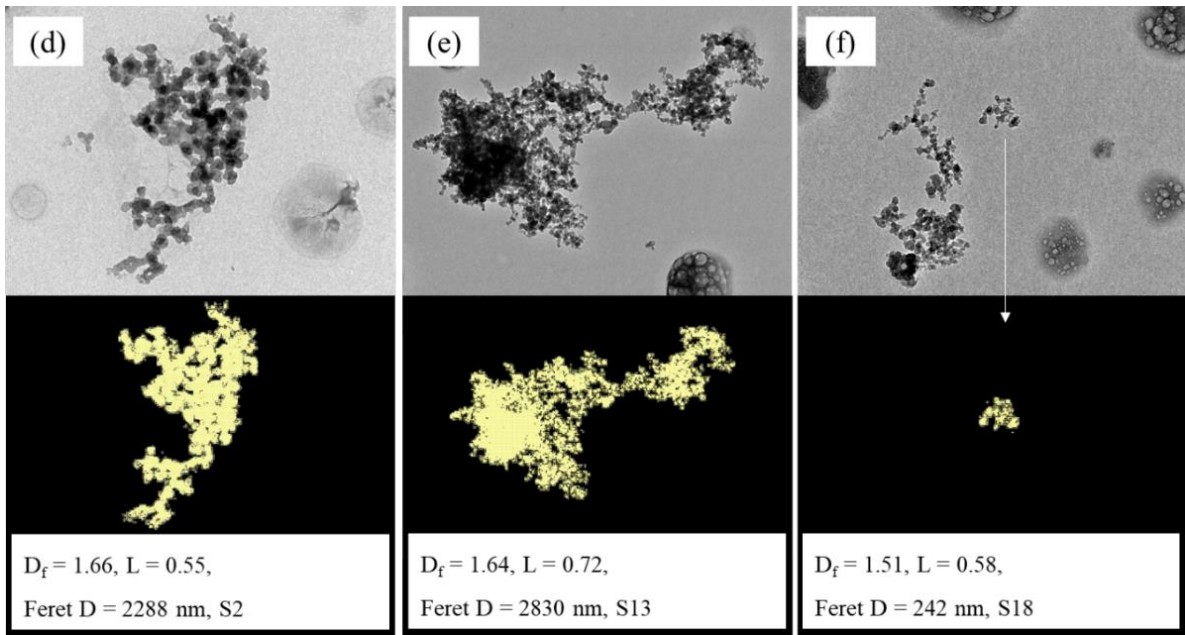


**Figure 6.** Examples of the BC TEM images and their corresponding Feret diameter (D), fractal dimension ($D_f$) and Lacunarity (L) based on the boxing counting


method from the fractal analysis: (a-c) BC particles collected during navigation and (d-f) during stop. More sampling information can be found in Table S1 of SI (serial


numbers N4, N7, N10, S2, S13, S18, etc.).


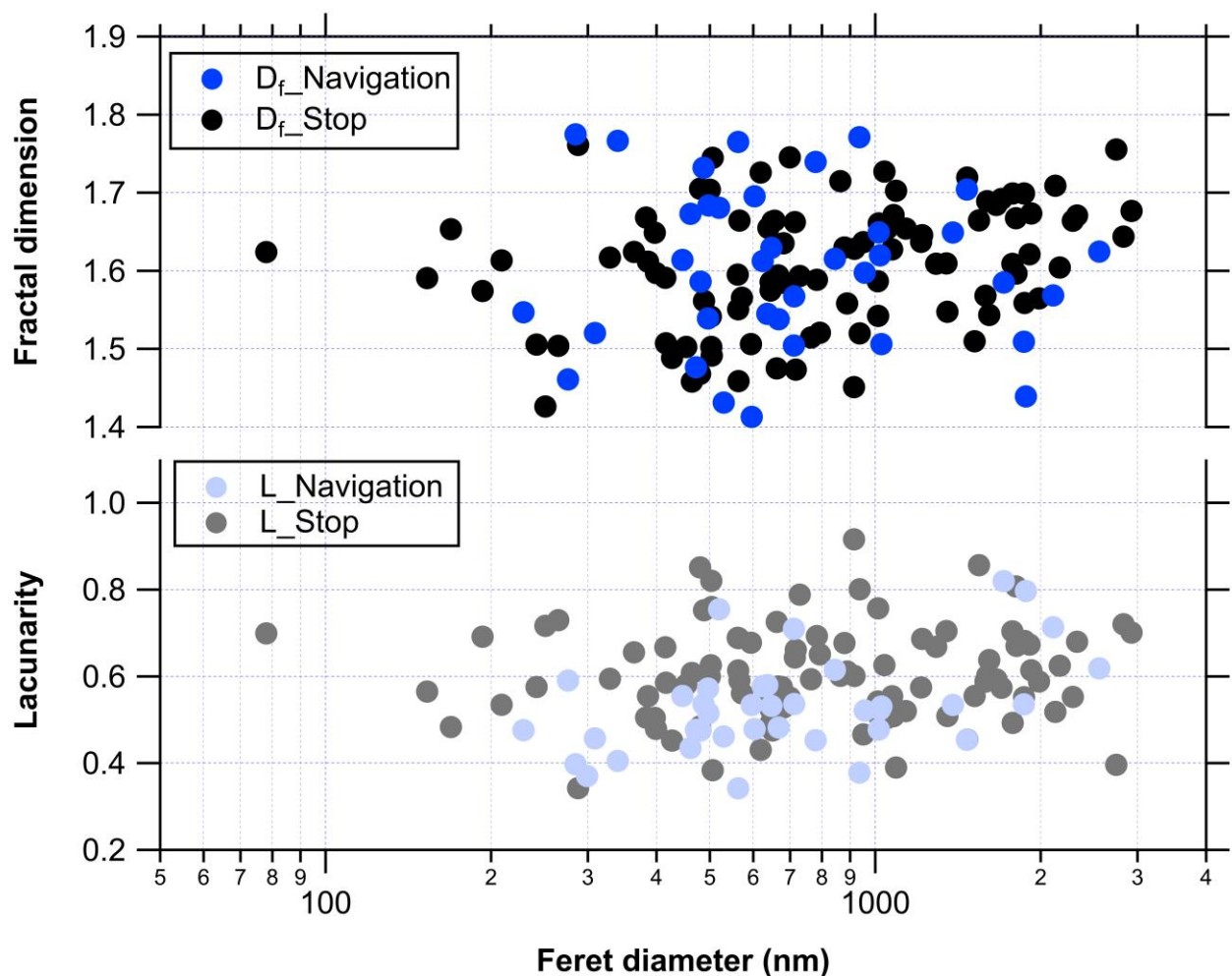


**Figure 7.** The size-dependent fractal dimension ($D_f$) and lacunarity (L) for each BC particle during navigation and stop. A total number of 134 data points are shown in
Figure 6.

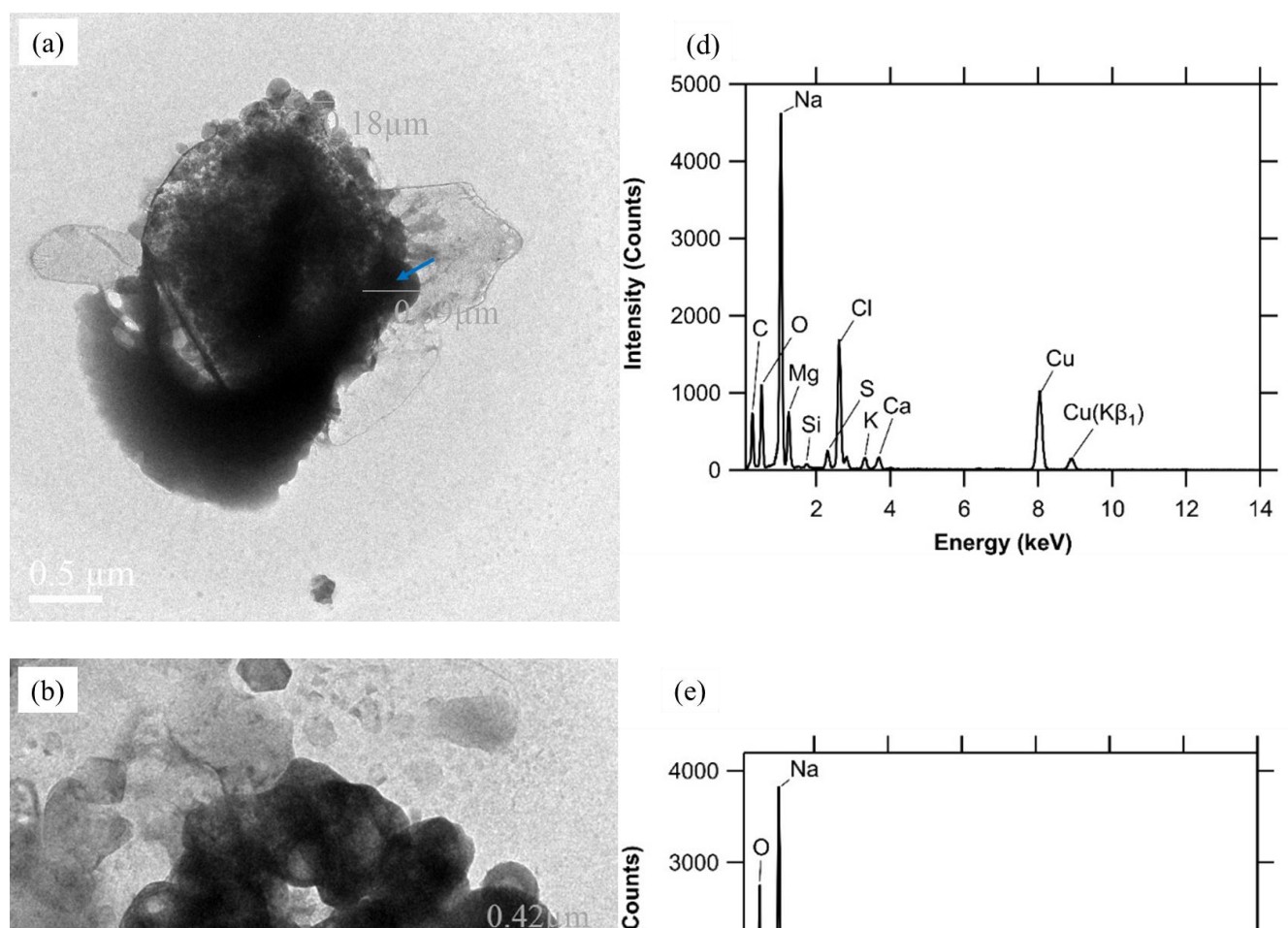



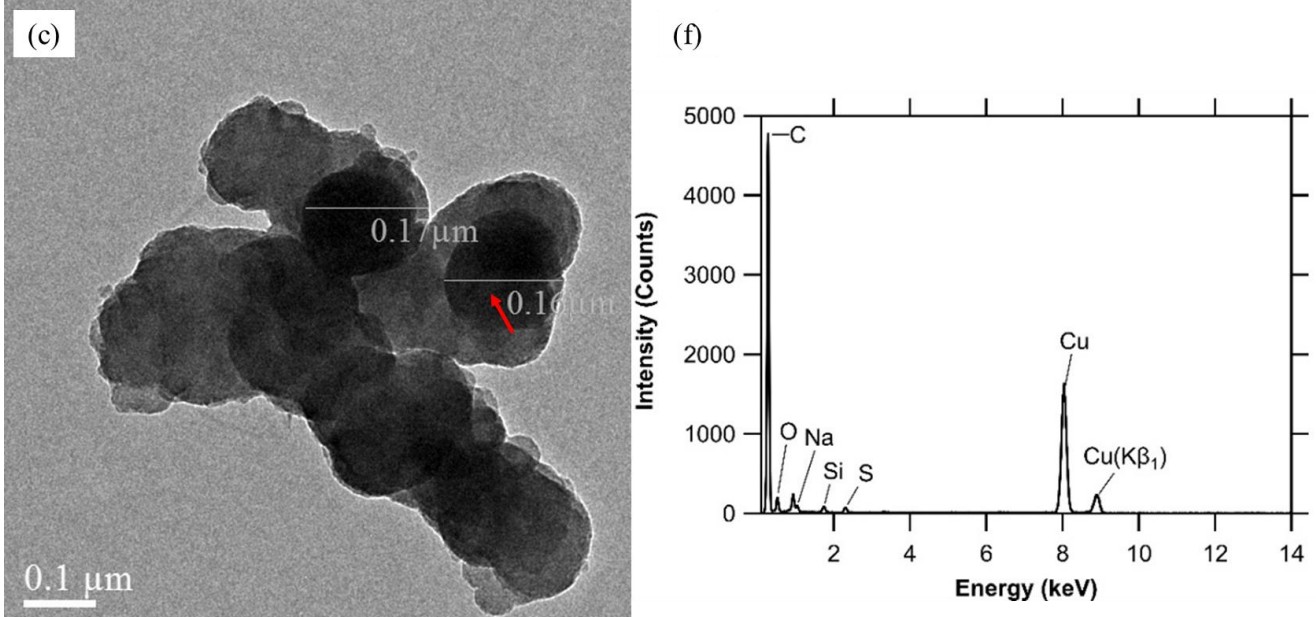


**Figure 8.** The example TEM images and their corresponding EDS spectra of particles collected during navigation: tar balls mixed with sea salt (a, d) at 8:55 on May 27, tar balls mixed with OC, sulfate (b, e), and amorphous carbon agglomerates mixed with sulfate (c, f) at 18:07 on Jun 01. The EDS spectra were collected by focusing the electron beam in the TEM and the illuminated area covers the center of the particle for elemental analysis. The blue arrows indicate tar balls and the red arrow indicates amorphous carbon. The EDS is obtained from beam focus on the center of the particle.

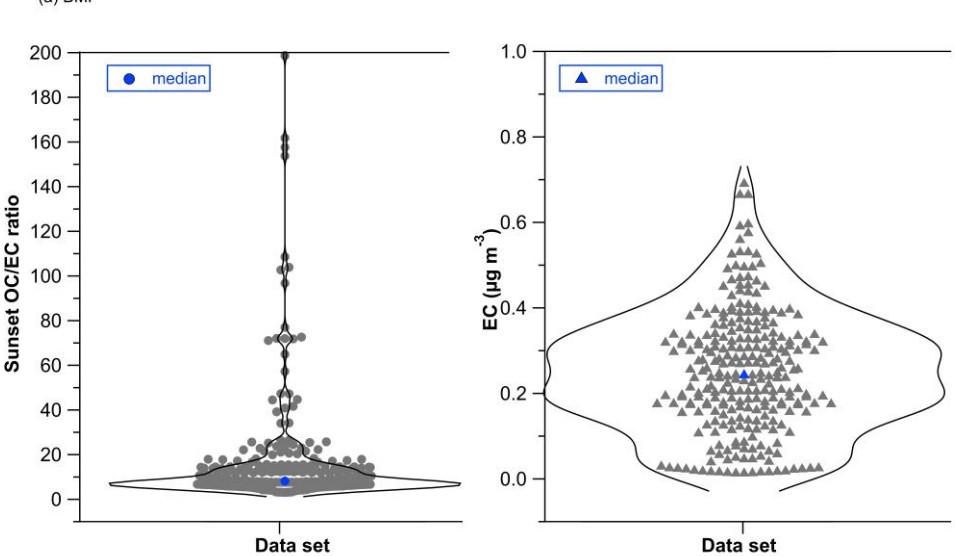


(b) TMP

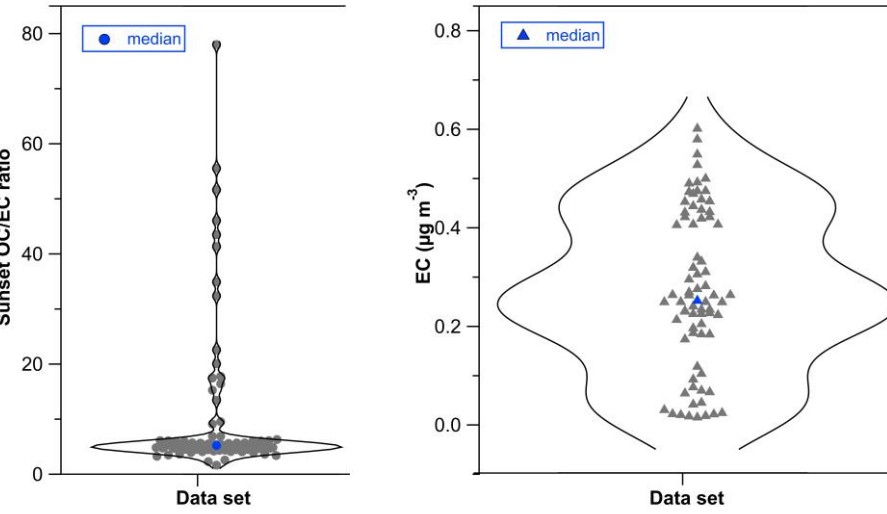


(c) AMP

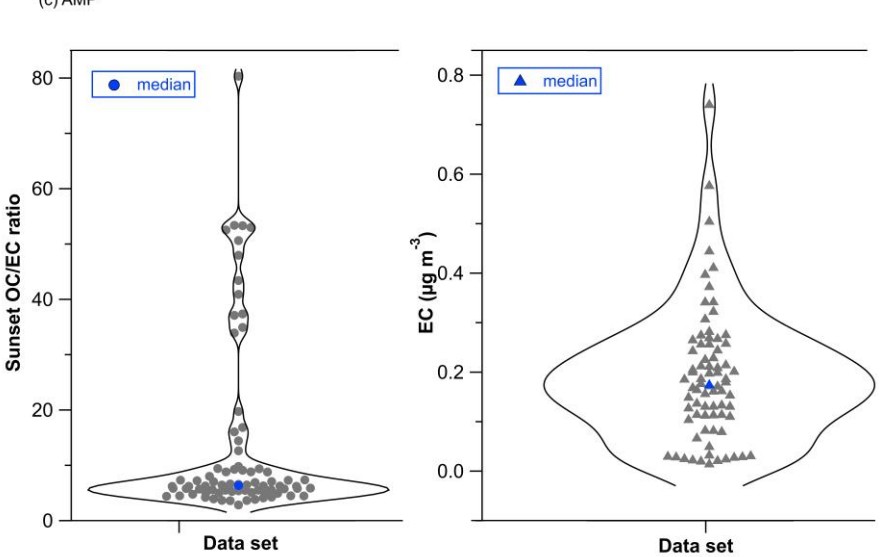


(d) SPP

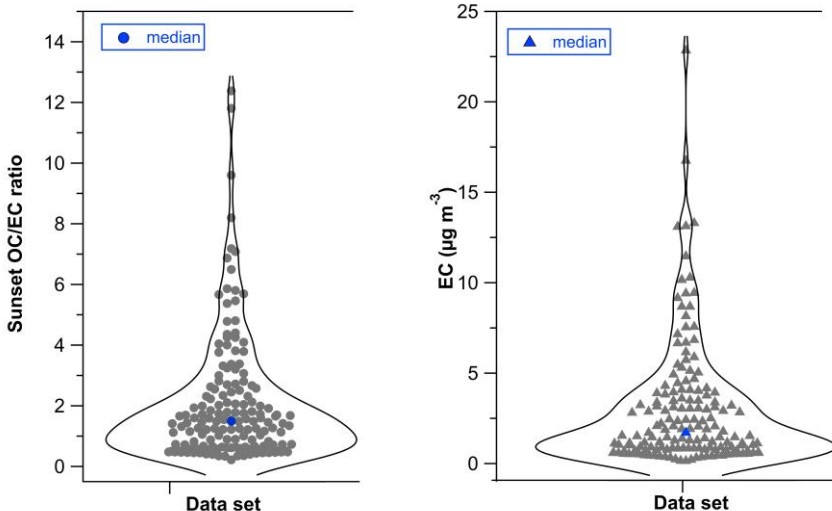


**Figure 9. Violin plots of the OC/EC ratios and EC concentrations for (a) before the monsoon period (BMP), (b) transition monsoon period (TMP), (c) after the monsoon**
**period (AMP), and (d) ship pollution period (SPP) based on the data from the Sunset OC/EC analyzer. The blue solid circles and triangles indicate median values of**
**Sunset OC/EC ratios and EC mass concentrations, respectively. The total data points are 551 in the data set, with a concentration range of 0.76-7.90 μg m$^{-3}$ for OC and**
**0.013-22.84 for EC μg m$^{-3}$. Particularly, all OC data is above LOD of 0.18 μg m$^{-3}$ while 29% of EC data is below LOD of 0.19 μg m$^{-3}$.**

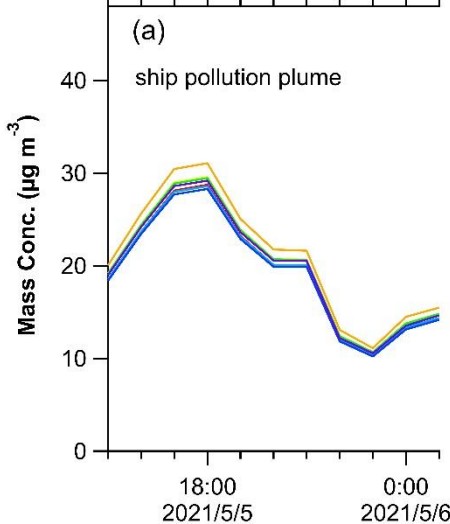
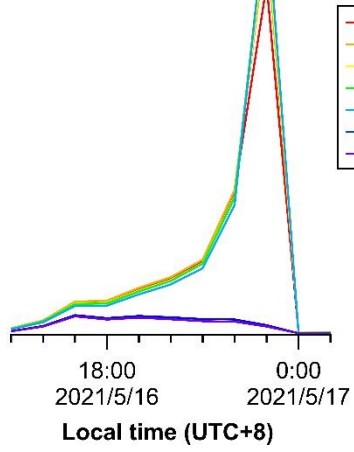
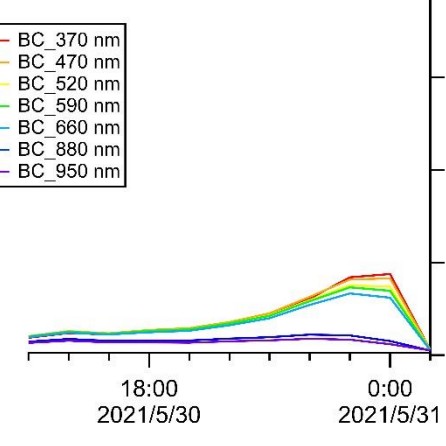


**Figure 10. The wavelength-dependent mass concentration from AE33 aethalometer for example spectra for (a) example of a ship pollution plume, and (b, c) two**
**significant biomass burning events during this campaign.**

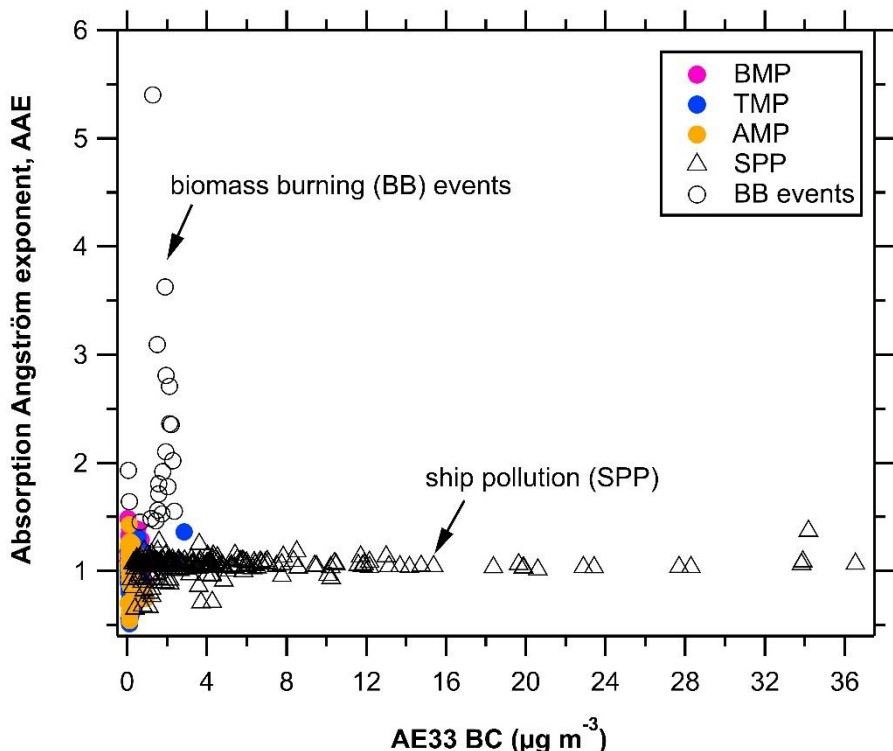


**Figure 11.** The absorption Angström exponent (AAE, all wavelengths) vs AE33 BC concentration for before monsoon period (BMP), transition monsoon period (TMP),
after monsoon period (AMP), and ship pollution period (SPP).

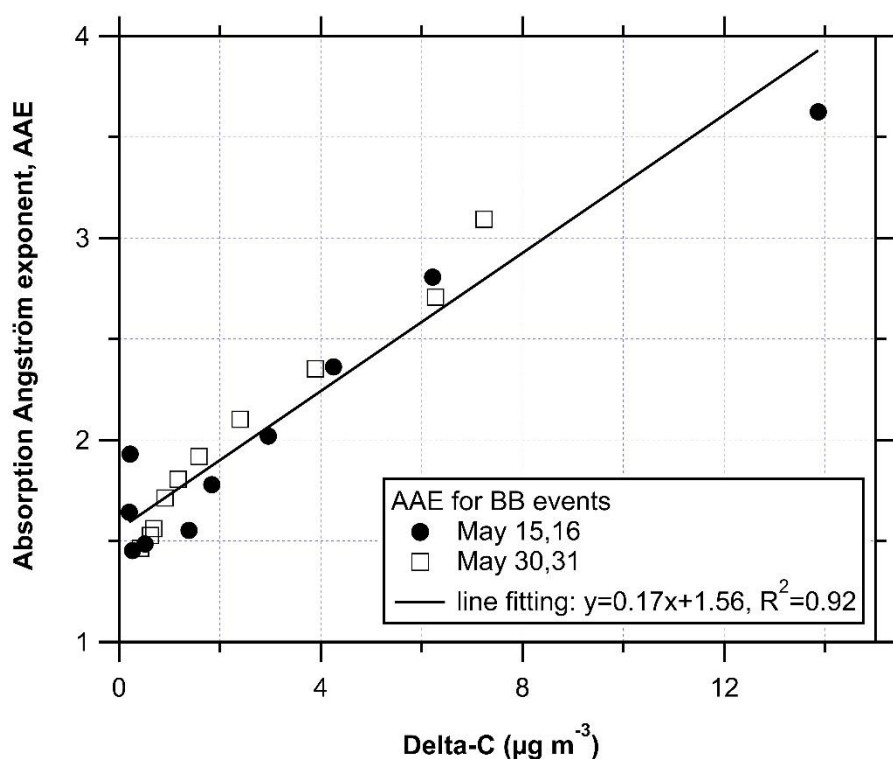


**Figure 12.** The absorption Angström exponent (AAE) vs the Delta-C concentration for the two biomass burning events: BB-1 at 6:00–7:00 on May 15 and 15:00–22:00

on May 16 during BMP, and BB-2 at 15:00-23:00 on May 30 and 00:00 on May 31 during TMP. The BB-1 and BB-2 data points are marked in solid circles and open
squares, respectively.

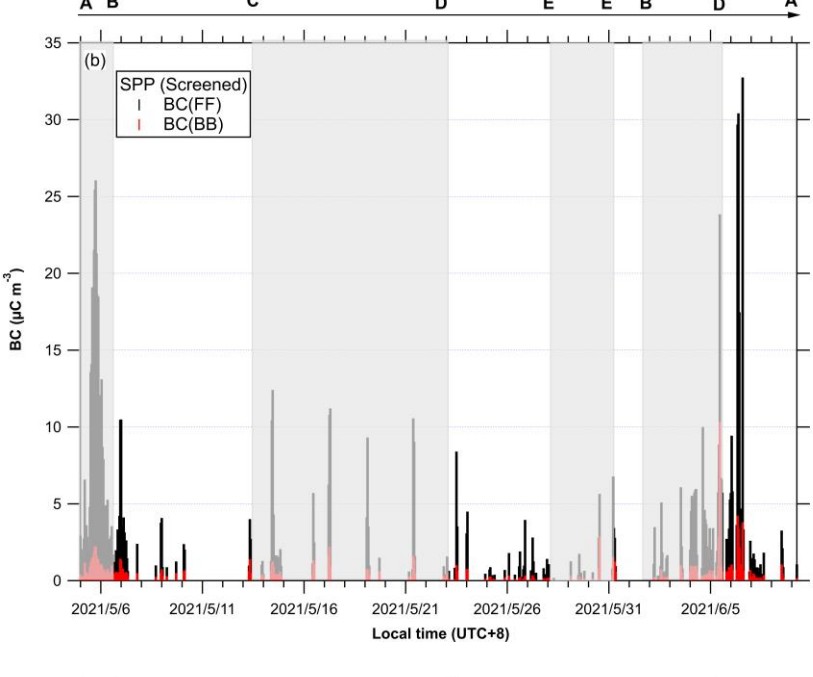


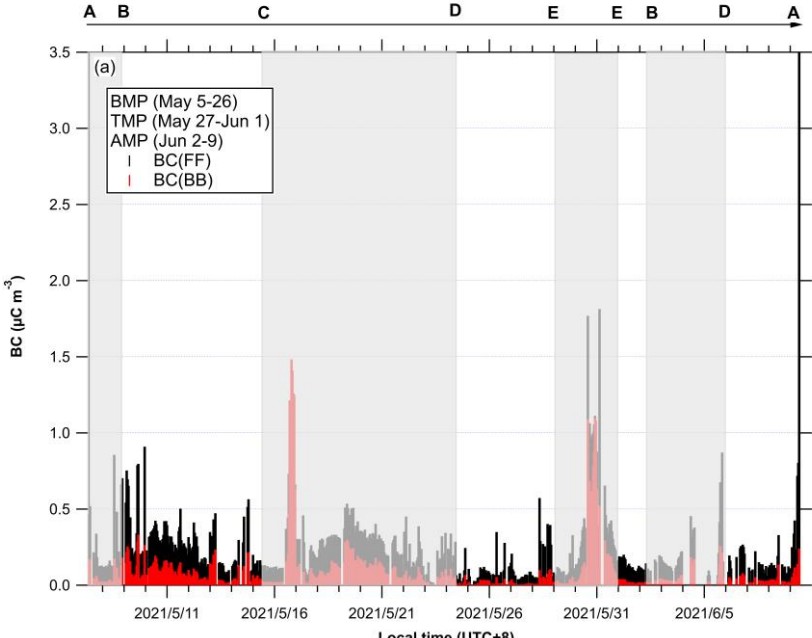

**Figure 13. Source apportionment of the BC particles using the two-component AAE model (AAE=1 for foil fuel (FF) and AAE=2 for biomass burning (BB): (a) before**
**monsoon period (BMP), transition monsoon period (TMP), after monsoon period (AMP), and (b) ship pollution period (SPP). The shaded and unshaded areas**
**sequentially indicate the cruise routes from AB, B to C, C to D, D to E, E to E (ship stop), E to B, B to D, and D to A, as marked in Figure 1.**