# Peer review of "Morphological and optical properties of carbonaceous aerosol particles from ship emissions and biomass burning during a summer cruise"

_EGUsphere, 2023_

## Author Comment (AC1)

**Response to Reviewer 1**

Comments:

The manuscript *Morphological and optical properties of carbonaceous aerosol particles from ship emissions and biomass burning during a summer cruise measurement in the South China Sea* investigated the morphological and absorption properties of BC particles in South China Sea and found that the size and mixing state of BC particles and tar balls differs during ship navigation and stop period, indicate the different aging degrees. Meanwhile, this study revealed biomass burning and fossil fuel combustion contributed respectively to 18–22% and 78–82% of all the BC light absorption, showed that biomass burning was predominantly from the Philippines and Southeast Asia before and after the summer monsoon during the cruise campaign. Generally, the study is interesting and meaningful. The study still needs some improvements. The manuscript needs some revision in order to be published:

We thank the reviewer for valuable comments and suggestions. We have revised the manuscript accordingly. All revised points are indicated in red in the manuscript. The point-by-point responses are given below. Note that we have rearranged the Results and Discussion section per the reviewer #2's suggestion.

1. What's the difference between Feret diameters and geometrical diameters?

Feret diameter and geometrical diameter are different. Feret diameter or Feret's diameter, is a measure of a particle size along a specific direction. We have revised the main text in lines 194-197, "In the analysis of particle size, the Feret diameter is defined as the distance between the parallel tangential lines that constrain the particle perpendicularly. In this study, we applied the Feret diameter as the longest distance between any two points along the boundary of the selected particles."

The term "geometrical diameter" signifies the distance between two points located on the surface of a geometric shape, with this line passing through the center of the shape. In this study, we utilized this concept in the analysis of transmission electron microscopy (TEM) images to quantify the size of tar balls with circular shapes. Specifically, we employed TEM data acquisition software to measure the geometrical diameters of the tar balls.

Why you use the former one to describe BC particles and use the latter to describe tar balls?

The Feret diameter is utilized to describe BC particles in this study because it allows for efficient particle counting capabilities before and after coating vaporization under electron beams using the ImageJ software.

The reason for using geometrical diameter is that bare tar balls are not found within the analyzed samples. Instead, tar balls mixed with other components (e.g., sea salt, organic matter, BC, and sulfate) were observed. Therefore, it is appropriate to quantify the size of the observed tar balls which excluded any coatings or additional materials using geometrical diameter.

We have revised the main text in lines 197-201, "Moreover, we utilized geometrical diameter to describe the size of tar balls with circular shape, which signifies the distance between two points located on the surface of a geometric shape, with this line passing through the center of the shape. The usage of geometrical diameter is reasonable for measuring the size of the observed tar balls which excluded any coatings or additional materials. Specifically, we employed TEM data acquisition software to measure the geometrical diameters of the observed tar balls."

The *Abstract* part is too long, maybe it will be better just listing the most important results in abstract.

The abstract has been revised according to the reviewer's suggestion. Specifically, the methodology section was condensed through the revision of lines 21-22, 26-27, 30 and the removal of the following sentences:

"Single particle samples were classified into two modes: "stop" when the ship was anchored and "navigation" when the ship sailed at high speed."

"The median OC/EC ratios were 8.14, 5.20, 6.35, and 2.63 during BMP, TMP, AMP, and SPP, respectively, showing higher OC/EC ratios for biomass burning emissions than for fossil fuel emissions. Additionally,"

"This study provides information about the morphology and the optical properties of carbonaceous aerosols which can be used to evaluate their effects on light absorption and hence the climatic radiative forcing in the SCS region."

2. In line 54, what's the meaning of onion-like graphite layer microstructures? From the TEM image, the BC particles don't look much like onions.

The term "onion-like" was originally used to describe the microstructure of nano-soot particles. To avoid the ambiguity, it has been revised to "graphene-like layers" and a reference has also been cited (Adachi et al., 2019) in line 48.

3.  As for Figure 6, why just chose some BC particles not all BC particles? Since not all

BC particles are included in the discussion, the conclusion that small-sized BCs are more easily encapsulated is not very convincing (In line 295).

We should point out that the particles in Figure 6 include pure BC and BC without thick coatings. We chose specific BC particles instead of all BC particles for two primary reasons: (1) A comprehensive investigation of all BC particles (BC-containing particles)

has recently been addressed by others (Pang et al., 2022). In their paper, all BC particles were discussed, including fresh soot, partly coated soot and embedded soot particles. It was found that the number fraction of embedded soot particles at the rural sites was higher, and these particles had the highest fractal dimension ($D_f$), implying that aged

BC particles became more compact after long-range transport. However, the characteristics of bare BC or pure BC exposed to other composition have not been comprehensively investigated. We hence focus on the pure or bare BC particles to explore their roles during aging in this study. (2) We convey that most aged BC particles were small after long-range transport, regardless they were initially small or became smaller due to the collapse of large BC aggregates. We have now revised the main text in lines 371-374, "Most BC particles were below 1 μm in Feret diameter during navigation (Figure 7), while their sizes cover a wide range below 3 μm during stop, implying that the aged BC particles become smaller after long-range transport. Despite only a total of 134 BC data points shown in Figure 7, the results are still statistically meaningful due to the wide range of BC sizes covered in our analysis. Note that the size change of a BC particle cannot be determined because the original size of the particle is unknown before the removal of the coatings."

In line 285, *among which were emitted from the own ship (e, f)*: Why only mention e/f, isn't d also from own ship's emissions?

The emissions from the own ship (research vessel) are much easily distinguished from other ships. For example, the BC particles (e, f in Figure 6) are emitted directly from the research vessel, showing the presence of large BC aggregates in the freshly emitted

BC particles. In contrast, aged BC particles (d in Figure 6) were thickly coated, which may originate from long-range transport of emissions from distant ships. We have now revised the relevant description in lines 354-359, "Comparatively, a mixture of aged

BC particles and much larger fresh BC particles as well as smaller scattered BC

particles during stop were found (Fig. 6d-f), which were likely emitted from other ships (Fig. 6d) and the research vessel (e, f). These TEM images showed that the compressed

BC particles are typically more aged and atmospherically processed, while the fractal

BC particles are fresh. Moreover, EDS analysis showed that sulfate formed from aqueous processes and less viscous organic coating indicate an aging process. Those

BC particles with Feret diameters larger than 2 μm during stop were fractal aggerates which could unlikely survive due to deposition during long-range transport."

4. In line 300,since Tar balls were frequently observed during the campaign, then what's the number fraction of tar balls in all particles?

We estimated the fraction of tar balls to be approximately 11.8% through the number of observed samples containing tar balls divided by the total number of analyzed samples, including both Navigation and Stop samples. We have now included this information in the main text in line 377, "Tar balls were frequently observed during the campaign with an estimated fraction of 11.8%."

5. There are mismatches between the appendix images and the image numbers mentioned in the main text: (1) Fig S5 is Map of the ship route, but line 287 says Fig S5

demonstrate "heavily coated internal BC particles were found during stop"; Fig S6 is titled "particles taken during navigation", but line 306 says Fig S6 contains tar ball mix with BC taken during stop. There are many more descriptions that don't match up.

We have thoroughly double checked and corrected all the mismatches/discrepancies both in the main text and the Supplementary Information (SI).

6. In line 335, *EC concentrations during SPP, ranging from 0.15 to 22.8 μg m$^{-3}$*, But the

EC concentration range for SPP in Fig 9 is around 1.7, why is that?

The EC concentrations ranged from *0.15 to 22.8 μg m$^{-3}$* with a median concentration of

1.7 μg m$^{-3}$ during SPP (Figure 9). We have revised Figure 9 and added more discussion in lines 399-402, "Compared with Figure 9d, the higher and more scattered OC/EC

ratios in Figure 9a, b, c are caused by the very low EC concentrations. The presence of extremely low EC concentrations, often falling below or close to the detection limit, can introduce discrepancies in the calculation of the OC/EC split, ultimately resulting in inaccurate EC concentrations (Bauer et al., 2009)."

**References**

Adachi, K., Sedlacek, A. J., III, Kleinman, L., Springston, S. R., Wang, J., Chand, D.,

Hubbe, J. M., Shilling, J. E., Onasch, T. B., Kinase, T., Sakata, K., Takahashi, Y., and Buseck, P. R.: Spherical tarball particles form through rapid chemical and physical changes of organic matter in biomass-burning smoke, Proc. Natl. Acad.

Sci. U.S.A., 116, 19336-19341, https://doi.org/10.1073/pnas.1900129116, 2019.

Bauer, J. J., Yu, X.-Y., Cary, R., Laulainen, N., and Berkowitz, C.: Characterization of the Sunset semi-continuous carbon aerosol analyzer, J. Air Waste Manag. Assoc.,

59, 826-833, https://doi.org/10.3155/1047-3289.59.7.826, 2009

Pang, Y., Wang, Y., Wang, Z., Zhang, Y., Liu, L., Kong, S., Liu, F., Shi, Z., and Li, W.:

Quantifying the fractal dimension and morphology of individual atmospheric soot aggregates, J. Geophys. Res. Atmos., 127, 10.1029/2021jd036055, 2022.

---

## Author Comment (AC2)

**Response to reviewer 2**

Comments:

This paper investigated the morphology and optical properties of carbonaceous aerosols collected during a ship cruise campaign. The results can help improve the knowledge gap related to ship emissions and aerosol above the ocean. However, there are still many places that need to be improved. Many points need to be better explained, and the manuscript needs to be better organized, making me have difficulty understanding and validating the results. Please see my comments below. I recommend a major revision.

We thank the reviewer for valuable comments and suggestions. We have revised the manuscript accordingly. All revised points are indicated in red in the manuscript. The point-by-point responses are given below.

Major comments:

1.       It is not very clear to me about your optical property measurements:

(1) For Aethalometer measurements, you must provide all necessary information, like data corrections. Aethalometer measures extinction, which is equal to absorption plus scattering. You should apply a correction for filter scattering based on your filter type.

Moreover, did you do any corrections for multi-scattering effects due to particle selves?

This can cause overestimations of absorption.

In this study, the absorption coefficient and BC concentrations were calculated according to the user's manual (page 30, manual version 1.54). Specifically, the absorption coefficients were calculated based on optical attenuation measurements at seven different wavelengths using a continuously loading filter in the employed

Aethalometer AE33 (Zhao et al., 2020; Yus-Díez et al., 2021). Here, data corrections were performed for the AE33, including several factors such as the multiple scattering parameters (C($\lambda$)=1.39 for the specific filter type used in the study), the leakage factor ($\zeta$ =0.01), and the compensation parameters ($K_{min}$=-0.005 and $K_{max}$=0.015). We have now added the above information in lines 152-154, "……through the filter tape (type

8060) at a sample flow rate of 5 L min$^{-1}$. Data corrections were made for the employed

Aethalometer AE33, considering the multiple scattering parameters (C($\lambda$)=1.39 for the used filter type), the leakage factor ($\zeta$=0.01), and the compensation parameters ($K_{min}$=-

0.005 and $K_{max}$=0.015).”

In addition, we have also included equations for the BC calculation (Eqs. 2-3) in lines

228-237.

(2) Moreover, some brown carbon can absorb at 880 nm, leading to overestimating BC

if you consider only BC absorbs at 880 nm. This can be improved by assuming

AAE_BC = 1 and applying fitting like babs(lambda)=a lamda^AAE_BC + b lamda^AAE_BrC for all wavelengths. Otherwise, you should call these BC equivalent

BC (eBC) since AE33 reports the equivalent mass of BC, which will absorb the same amount of light at that wavelength.

We agree with the reviewer that the BC derived from AE33 in this study should be denoted as equivalent BC (eBC) (Yus-Díez et al., 2021). We have now added a sentence to reflect the change of the notation in lines 149-150, “Note that the BC mass concentrations derived from AE33 are referred to as equivalent BC mass concentrations due to the light absorption of both BC and BrC at 880 nm.”

(3) It is not very clear to me how you measure optical EC. Could you provide more details about Sunset optical EC calculation? Does it use the same method as AE33?

Since BrC might still significantly absorb at 660 nm (Cheng et al., 2019; Corbin et al.,

2019) and you might not be able to correct multi-scattering, filter scattering, and loading effects related to filter-based optical measurements, it is essential to discuss your method. This is also related to OC/EC analysis since pyrolysis EC correction is based on transmission and reflection of 660 nm wavelength. Thus, OC/EC analysis typically overestimates EC (Cheng et al., 2019).

Optical EC concentrations are measured in the Sunset OC/EC analyzer based on the transmission of 660 nm wavelength light through the quartz fiber filter employed for sampling, similar to the AE33 for optical BC measurements. Optical EC is defined as the apparent EC on the filter based on a fixed absorption coefficient and the apparent absorbance. The absorption coefficient is applied according to the user's manual of the

Sunset OC/EC (Page 59-61). Both our study and a previous study (Brown et al., 2019)

showed that the optical EC concentrations (from Sunset) were comparable with the BC concentrations (from AE33). We admit that the resultant optical EC concentrations from the instrument output may be overestimated due to the limitation of the filter-based optical measurements. We have now revised the text considering the above information in lines 176-182, "The Sunset OC/EC analyzer also measures optical EC based on the transmission of 660 nm wavelength light through the quartz fiber filter employed for sampling, similar to the AE33 for optical BC measurements. Optical EC is defined as the apparent EC on the filter based on the measured apparent absorbance and the fixed absorption coefficient according to the user's manual of the Sunset OC/EC. Both our study and a previous study (Brown et al., 2019) showed that the optical EC concentrations from Sunset were comparable with the BC concentrations from AE33. Note that the resultant optical EC concentrations from the instrument output may be overestimated due to the limitation of the filter-based optical measurements."

(4) Also, did you convert measured OC and EC to organic and black carbon mass since the OC-EC analyzer reports carbon mass in organic and BC, which will be smaller than organic and BC mass due to excluding other elements like oxygen and nitrogen?

No, we did not. We only report the element carbon mass.

The Sunset OC/EC analyzer uses a modified NIOSH 5040 thermal-optical protocol as its default protocol. This protocol provides a relatively reliable determination of OC, EC, and the OCEC split. The thermal-optical protocol first evolves OC in pure helium (He), which is carried into a manganese dioxide oxidizing oven for conversion to carbon dioxide ($CO_2$). The $CO_2$ is then quantified by determining its absorbance directly using a tunable red diode laser in a self-contained flow through non-dispersive infrared (NDIR) detector as it exits the oxidizing oven by the He carrier gas. EC is then desorbed in an oxygen ($O_2$) blend carrier gas and quantified in the same way as OC. At the end of each run, an internal standard of known volume of methane ($CH_4$) is injected and oxidized to $CO_2$ to ensure accurate quantification of OC and EC. Therefore, the OC and EC concentrations only contain the element carbon of the organic matters and the BC mass.

(5) It needs to be clarified which method you used for the AAE discussion in your paper. Also, the details of your AAE model need to be included, which makes me unable to understand your results. Moreover, for your AAE model, how did you decide on AAE values of 1 and 2 for FF and BB? I think these values are too low, and I suggest using a range instead of 1 value to account for the uncertainties.

We directly followed the AAE model in the user's manual, using AAE values of 1 and 2 for FF and BB, respectively. Details of the AAE model and two methods for AAE calculation are included in the main text in lines 238-253.

The measured median AAE values for the classified periods (BMP, TMP, AMP, and SPP) ranged from 1.02-1.14 and 1.85-1.86 for two significant biomass events, which are very close to 1 and 2 for FF and BB, respectively.

Why do you have AAE values below 1 in Figure 11? Are these noises due to low absorbing particle loading?

The AAE values below 1 shown in Figure 11 are not from noises. To avoid ambiguity, we have added one sentence in the revision in lines 440-441, "The AAE values below 1 in Figure 11 are not noises, in some cases due to aerosols from fossil fuel (Ezani et al., 2021) and in other cases, they can be even lower than 0.5 when paired with wavelengths of 470 and 660 nm (Laing et al., 2020)."

2. It is also not very clear to me in some single particle analyses:

(1) Do you measure max Feret or mean Feret diameter or Feret diameter measured at an angle of 90 degrees to max Feret diameter? How many BC particles have you analyzed? I also did not see the details about your $D_f$ and lacunarity calculation.

In this study, as illustrated in Figure 5, we measured maximum Feret diameters for a total of 15,624 particles from a total of 34 representative examples, and this number is included in the text in lines 193-194. Among them, we selected 134 BC particles for the maximum Feret diameter and fractal analysis on pure BC particles or BC residue as shown in Figure 7. Here, we employed the ImageJ software to calculate $D_f$ and lacunarity using the Fraclac plugin. A detailed description of $D_f$ and lacunarity calculations are included in Section 2 of the SI.

(2) For your TEM imaging, I am very surprised that all organics can be evaporated at
only 120 kV after beam focusing since the evaporation should occur during the vacuum
process, and beam damage is typically not like this (typically for sulfate, and you will
see some residual as the empty frame). I never see coating removed that completely,
even with 300 kV acceleration voltage. It only happens during heating TEM
experiments by heating the substrate to a few hundred ℃. Did you do EDX mapping
on these particles to see spatial distribution in the particles? It will be helpful to
determine particle types based on both shape and elemental composition. Your EDX
spectrum only shows a few positions, which might not represent the whole particle.

We agree with the reviewer that organic coatings cannot be completely removed under
the electron beam with an acceleration voltage of 120 kV. From the example images
shown in the main text (Figure 6c) and the SI (Figure S7), we can see significant
residues of particles after beam focus with the acceleration voltage of 120 kV.
Unfortunately, we could not perform EDX mapping to get the shape and composition
for individual particle due to the limitation of the TEM instrument employed in this
study. Instead, we obtained the EDS spectra by focusing the beam on the center of the
particle. We should point out that a 120 kV accelerating electron beam may be
sufficiently powerful for the analysis of aerosol particles in TEM, as supported by
Adachi et al. (2017).

We notice that no significant coatings remained as shown in Figure 6a, b, e and f for
the BC fractals. However, these BC particles contained thin coatings because they are
collected from very fresh emissions of the own ship during ship stop or of other ships
during navigation. We have now revised the caption of Figure 8 in lines 915-918, "The
EDS spectra were collected by focusing the electron beam in the TEM and the
illuminated area covers the center of the particle for elemental analysis."

For tar ball particles, did you observe individual tar balls and tar ball aggregates (see
Girotto et al., 2018)? Did you take tilted view images to confirm these round particles
are spherical since they might not be domelike and flat (see Cheng et al., 2021)? Could
you estimate the number fraction of tar balls in the samples?

In this study, we did not observe individual tar balls but only tar balls mixed with other components. When taking the TEM images, we did tilt the sample holder at an angle of 25° for thorough observation. We estimate an approximately 11.8% of tar balls in the samples. We have now added the relevant information in line 145, "The substrate holder of TEM was tilted 25° for thorough inspection during imaging and EDS analysis." and in lines 377-380 in the revision, "Tar balls were frequently observed during the campaign with an estimated sample fraction of about 11.8%. Fractal-like tar ball aggregates were usually found in wildfire smokes (Girotto et al., 2018); however, in this study, spherical tar ball particles were observed in the marine atmosphere and were mixed with sea salt (Fig. 8a and d for TEM image and EDS spectrum, respectively), organic carbon and sulfate (Fig. 8b and e) from the samples collected on May 27 during navigation."

(3) Could you add more discussion on how you determine aging and fresh particles based on TEM images? Compressed BC is typically more aged and atmospherically processed, and fractal soot is fresh. Moreover, sulfate (aqueous processing) and less viscous organic coating can be indicators of aging. Did you observe this difference in your navigation and stop cases? Moreover, you should observe bimodal distribution in stop cases.

We agree with the reviewer regarding the differences between aged and fresh BC. We have now added more discussion to reflect the reviewer's points in the revision (lines 354-359), "Comparatively, a mixture of aged BC particles and much larger fresh BC particles as well as smaller scattered BC particles during stop were found (Fig. 6d-f), which were likely emitted from other ships (Fig. 6d) and the research vessel (e, f). These TEM images showed that the compressed BC particles are typically more aged and atmospherically processed, while the fractal BC particles are fresh. Moreover, EDS analysis showed that sulfate formed from aqueous processes and less viscous organic coating indicate an aging process. Those BC particles with Feret diameters larger than 2 μm during stop were fractal aggerates which could unlikely survive due to deposition during long-range transport."

We also agree with the reviewer that a bimodal distribution should be observed during stop. However, we couldn't successfully obtain a bimodal or multi-peak fit for the data of the stop cases using multi-peak fitting function in the Igor Pro software, as shown in Figure S6. We believe that single peak fitting best described the distribution in our stop cases, as illustrated in Figure 5. To clarify this point, we have added sentences in lines 335-337 in the revision, "Note that we could not successfully obtain a bimodal or multi-peak fit for the data of the stop cases using multi-peak fitting function in the Igor Pro software, as shown in Figure S6. Hence, we believe that single peak fitting best described the distribution in our stop cases, as illustrated in Figure 5." and have included Figure S6 in the SI.

3.      I got lost in the different classifications of your samples. Why don't you use the same classification? Moreover, the classification for the campaign period should not class SPP as an independent period since it is a subset of others.

Here, we classified the samples according to both temporal and spatial variations during the campaign. For online sampling, we focused on the differences between local emissions and long-range transport sources. For offline single particle analysis using TEM, we then focused on the influence during ship stop and navigation. We classified SPP as a special period since it could provide meaningful comparisons of fresh ship (research vessel) emissions with other scenarios and cases in term of the light absorption properties. Hence, we think the classification is appropriate and reasonable. We have now revised the text in lines 323-325, "SPP (ship pollution period), ~35% of the online measurement data could be attributed to this category in this study due to the interference from the research vessel own emissions."

4.      I suggest adding a table in either the main text or SI to show the thresholds you used to identify different sources,

Per the reviewer's suggestion, we have included Table 1 which outlines the classification of observation periods and the wind directions. We believe that it can serve as a reference for the thresholds to identify different sources. We have included the text in lines 318-325 to reflect the changes, "Here, we classified the campaign period into several groups based on the cruise route, change of wind direction during monsoon, backward trajectories, and ship pollution, as listed in Table 1: (1) BMP-1 (before monsoon period 1), AB route mainly with northeast wind direction during May 05–09; (2) BMP-2, B→C→D route close to the Philippines primarily with southeast wind direction during May 10–22; (3) BMP-3, D→E close to mainland China with the same wind direction as BMP-2 during May 23–26; (4) TMP (transition monsoon period), EB route with south wind direction during May 27–Jun 01; (5) AMP (after monsoon period), B→D→A route with southwest wind direction during June 02–09; (6) SPP (ship pollution period), ~35% of the online measurement data could be attributed to this category in this study due to the interference from the research vessel own emissions."

Your figure numbers in the main text should be checked carefully since some places refer to wrong figures.

We have thoroughly checked the figure numbers in the revision and the SI.

Specific comments:

1.      L50-51, "Carbonaceous aerosols … 2020)." BrC is a special subset of OC, so it should not be parallel with OC and BC.

The sentence has been revised by removing "and brown carbon (BrC)" in the revision (lines 45-46), "Carbonaceous aerosols (e.g., organic carbon (OC), elemental carbon (EC)/black carbon (BC)) profoundly impact regional and global climate (Corbin et al., 2019; Lu et al., 2020; Rabha and Saikia, 2020)."

2.      L59-61, "BrC typically … respectively)." This is not true. BC should have a higher imaginary part or MAC from Visible to NIR-IR than BrC.

These sentences have been removed in the revision.

3.      L64-66, "These particles … 2005)." Tar balls belong to BrC because they are light absorbing organic.

This sentence has been modified in the revision (lines 57-58), "These particles also belong to BrC because they are light-absorbing organics (Adachi et al., 2019; Hand et al., 2005)."

4.  L87-88, "When BC … 2021)." Well internally mixed means different species are homogeneously distributed inside a particle, which is impossible for BC and other materials. Also, the shielding and lensing effects should depend on the coating thickness (Lack and Cappa, 2010).

These related sentences have been revised according to the reviewer's suggestion (lines 69-72), "The shielding and lensing effects depend on the coating thickness over BC (Lack and Cappa, 2010). When BC is well internally mixed with BrC, its total absorption enhancement becomes smaller than the enhancements of not well mixed counterparts due to the absorptive coating that acts as a shield (Feng et al., 2021). Moreover, it is impossible for BC and other materials to be homogeneously distributed."

5.  DKL-2 should be a two-stage cascade impactor. What is the cut-off size for the other stage? Are there any references to validate the cut-off size? Section S1 is not necessary if someone has already published these results. Moreover, Section S1 is a theoretical calculation. Did you test the cut-off size? Did you only collect on stage with 50% cut-off = 0.2 μm? Why did it not include the other stage?

The sampler (DKL-2) employed in this study is a single-stage cascade impactor, capable of collecting either fine or coarse particles by a 0.3 mm or 0.5 mm diameter nozzle, respectively. It can be utilized with one stage (either fine or coarse particles) at any given time, which is different from the two-stage cascade impactor (Adachi et al., 2017). The sampler was utilized in previous studies without mentioning the validation of the cut-off size (Chen et al., 2023; Dong et al., 2018) and hence we included a theoretical cut-size estimation in Section 1 of the SI. However, we did not carry out experiments to test the calculated cut-off sizes. Here, we collect fine particles with a 0.3 mm nozzle for the analysis of BC particles, obtaining a calculated 50% cut-off size at 0.2 μm. To clarify this, we have now revised sentences in lines 133-138 to include the reviewer's suggestions, "Single particles were collected on the TEM grids (3.05 mm I.D., copper meshed and covered with lacey carbon film) located on the front deck during ship navigation and stop using a single-stage particle sampler (DKL-2, Genstar Electronic Technology Co., Ltd., China) which is the same as other studies (Chen et al.,

2023; Dong et al., 2018; Liu et al., 2021; Pang et al., 2022). The sampling flow rate and time were set at 1 L min$^{-1}$ and 10 min, respectively, for each collection. The nozzle diameter of this single-cascade impactor is 0.3 mm. The particles with aerodynamic diameters above 0.2 μm were collected with a collection efficiency of 50%, assuming a particle density of 1.5 kg m$^{-3}$ (Marple and Olson, 2011)."

6.      L153-154, "The BC mass … time resolution." I do not think AE33 has a time resolution of 1 second.

We used one minute time resolution in this study for the AE33 measurements. We have modified the text in the revision (lines 148-149), "The BC mass concentrations were measured by an aethalometer (Model AE33, Magee Scientific, USA) with a time resolution of one minute."

7.      L176-177, "Here, … campaign." How do you determine this value? These should be instrumenting noise or contamination, not your detection limit. You should use a standard with a known concentration to calibrate the detection limit.

We agree with the reviewer that the three standard deviation of those blank measurements should correspond to instrument noises or contaminations rather than the instrument detection limit. We have now revised the sentence in lines 173-176 in the revision, "Here, we estimated the instrument noises (including contamination) of 0.15, and 0.012 μg m$^{-3}$ for OC and EC based on 26 effective blank measurements with 3

times the standard deviation (3σ) during the campaign. The limit of detetion (LOD) for

OC and EC is 0.18 and 0.19 μg m$^{-3}$, respectively, calculated as three times the standard deviation of replicate measurements of a standard sucrose solution with a carbon content of 10.516 μg m$^{-3}$. "

8.      L178-180, "The measurements … the ship." Do you have any references for these instruments? What is the time resolution? What are their uncertainties?

Per the reviewer's suggestions, we have now included the relevant information in lines

184-188 in the revision, "The measurements of solar radiation (SR), temperature (T), pressure (P), relative humidity (RH), relative wind direction (RWD), and relative wind speed (RWS) were provided by the automatic weather station (AWS430, Vaisala Inc.,

Finland) (Song et al., 2022) equipped on the front deck of the research vessel. This station comprises a range of integrated sensors, including a wind speed and direction sensor (model WMT702), a temperature and humidity sensor (model HMP155), and an atmospheric pressure sensor (model BARO-1). The cruise route for ship navigation is from the global positioning system (GPS) onboard the ship (Seapath 330+, Kongsberg Inc., Norway)."

Detailed information of the time resolution and accuracy is included in section in the SI (Section 11), "The time resolutions for the meteorological and GPS data are 3 seconds. The position accuracies for X and Y axes are 1 cm +1 ppm RMS (root mean square), and for Z axis is 2 cm +1 ppm RMS. The accuracy of wind speed and wind direction is $\pm$ 0.2 m s$^{-1}$ (or 3% of reading) and $\pm$ 2°, respectively. The accuracy of temperature at 20–60 °C is $\pm$ (0.07 + 0.0025 × temperature) °C. The accuracy of relative humidity at -20 to + 40 °C is $\pm$ (1 + 0.008 ×reading) %RH. The accuracy of pressure with the factory calibration is $\pm$ 0.15 hPa (Class A)."

9.      L197-198, "The navigation … TEM samples)." Is the relative wind direction relative to the North or ship direction? How did you determine the criteria for wind speed and direction?

The reviewer is correct. The wind direction is referenced to the North, whereas the relative wind direction is aligned with the ship's orientation. The automatic weather station provides data such as ship heading (orientation), the true wind speed/direction, and the relative wind speed/direction. The relative wind speed and direction are converted by vector calculation. We have now included information on the relative wind direction/speed in lines 217-220 in the revision, "The wind direction (speed) and relative wind direction (speed) are calculated by Eq. (1) (Aijjou et al., 2020).

$$V_R = \sqrt{V_s^2 + V_w^2 + 2 * V_s * V_w * \cos \alpha} \tag{1}$$

where $V_R$ is the relative wind direction (speed), $V_s$ is the ship direction (speed), $V_w$ is the true wind direction (speed), $\alpha$ is the angle between the ship heading and the true wind direction."

10.    L203-204, "Here, we … transport." Could you provide details about how did you distinguish these? Based on chemical composition? Other ship emissions might not be easy to separate from your ship emission.

We agree with the reviewer that other ship emissions might not be easy to separate from the own ship emission and we employ the following criteria, which have been included in lines 223-226 in the revision, "Here, we distinguished the own ship emissions (research vessel) from those of other ships or long-range transport based on the following criteria: low relative wind speed (< 5 m s$^{-1}$), relative wind direction encompassing ship exhaust ( 80–280°), and a substantial AE33-derived hourly averaged BC mass concentration (>2 μg m$^{-3}$). Other ship emissions far from the research vessel are treated as a part of the transported air masses in this study."

L207-208, "Here, we … variations." I suggest using a subscript to indicate BC mass from OC-EC or AE33. It is unclear to me.

In this study, the BC data obtained from the AE33 are referred to as BC, while data from the OC/EC analyzer are denoted as thermal OC, thermal EC, and optical EC. The optical EC is not extensively discussed and does not play a critical role in our analysis. Therefore, we believe that the employed descriptive names should provide enough clarity.

To avoid the ambiguity, we have now revised the text in lines 228-229, "In this study, BC data obtained from the AE33 are referred to as BC, while data from the OC/EC analyzer are denoted as thermal OC, thermal EC, and optical EC."

11.    Figure 1. The color bar needs to be clarified. I suggest using colors with higher color resolution.

The color resolution of Figures 1, 4, and S3 has been upgraded.

12.    L250-252, "It should be …. html.en." It is not shown as an increase in wind speed and RH and a decrease in pressure in Figure 2 for the typhoon period. Could you explain that? Moreover, I suggest adding a SI figure to show the typhoon.

We thank the reviewer for pointing this out. We have included a map for the typhoon track and a chart for the central pressure of the typhoon in the SI (Section 5, Figure S5).

In addition, we have now addressed the reviewer's concern in lines 292-297 in the revision, "The typhoon was initiated at 02:00 local time on May 31 and dissipated at 14:00 on June 05, 2021 (Figure S5). It passed over our cruise route from June 03 to June 05, 2021. While no significant increase of absolute wind speed was seen in Figure 2, a significant increase of relative wind speed was shown in Figure S2, along with an obvious decrease of atmospheric pressure during the typhoon period (Figure S5). The measured relative humidity increased from May 27 to June 01, prior to the presence of the typhoon, which can be attributed to the decrease of ambient temperature during this period."

13. L253-257, "Figure 3 … 80-280º". I expect a detailed discussion of Figure 3 since that tells lots of important information. Why do you see more BC after the monsoon, which I expect pollution will be removed by rain? Also, why do you see more BC before May 8th? Or OC, did you observe any diurnal trend or other trend? I suggest labeling the sampling period and path in Figures 2 and 3 by adding shaded areas—same suggestion for all other time serial figures.

We agree with the reviewer. We have now updated Figures 2, 3, 13, and S2 with shaded areas. In addition, we have added more discussion on Figure 3 in lines 303-307 in the revision, "Before May 08 and after June 05, higher UVPM, OC, and EC concentrations were observed, which can be attributed to significant fresh ship emissions from the research vessel, as evidenced by simultaneous higher BC concentrations. Similar spikes in BC concentrations were observed during other measurement periods, either preceding or following the monsoon period, which were caused by emissions from the frequent stops and starts of the ship. Note that no significant diurnal trend for OC was observed during those aforementioned periods."

14. L262-264, "The choice of … 2007)." You should adjust the bin width to make the distance between each bin is constant in log scale. I suggest using same bin size to help reader visualize easily.

We have included a sentence to clarify this point in lines 329-330 in the revision, "The distribution is represented with histograms starting at 50 nm, a width interval of 20 nm, and a bin number of 200."

15. L275-277, "The BC … 2020a)." It is hard to see the coating in a and c. Both look like embedded to me. Do you have better images?

Figure 6a shows a typical embedded type, while a core-shell type in Figure 5c. Please refer to lines 345-347 in the revision, "The BC particles collected during navigation are in the embedded (a), external (b), or core-shell (c) mixing states classified with the methods which are based on single particle analysis of island and mountain samples across East China Sea and Japan (Adachi et al., 2014; Sun et al., 2020a)." In addition, we have also included images captured just before and after electron focus in the SI (Section 7, Figures S7c and S7f).

16. L284-285, "Comparatively … (e,f)." How did you know this? This is not clear to me.

We have now added more description in lines 354-361 in the revision, as have been addressed in the above main question #1 (3), "Comparatively, a mixture of aged BC particles and much larger fresh BC particles as well as smaller scattered BC particles during stop were found (Fig. 6d-f), which were likely emitted from other ships (Fig. 6d) and the research vessel (e, f). These TEM images showed that the compressed BC particles are typically more aged and atmospherically processed, while the fractal BC particles are fresh. Moreover, EDS analysis showed that sulfate formed from aqueous processes and less viscous organic coating indicate an aging process."

17. L286-287, "In addition, … (Fig. S5)." They could also be condensation of organic during cooling after emitted from engine if you do not see them spread out (high viscous).

We agree with the reviewer that the particles could also be condensation of organic from engine emissions. We have now added more discussion in lines 359-361 in the revision, "In addition, heavily coated internal BC particles were found during stop due to the mixing between ship pollution and the marine air (Fig. S9). Moreover, such particles could also be condensation of organics during the cooling process after they were emitted from the ship engine."

18.    Figure 6: Does Fig 6 just show results from a portion of BC you imaged? If yes, why don't you show all of them? Do you think your results is statistically significant since your sample number is very low.

We have now added more data points in Figure 7 to show all the BC particles (a total of 134) from the 34 TEM grid samples observed by the TEM. Similar results were obtained from other particles collected on a distant island sampling in East China Sea (Figure S16). We have now added a sentence in lines 372-373 in the revision, "Despite a total of 134 BC data points shown in Figure 7, the results are still statistically meaningful due to the wide range of BC sizes covered in our analysis."

L294-295, "Figure 6 … during transport." This is unclear to me. Please explain this in detail. Did you observe smaller particles have more coating? If yes, have you tried to quantify the size change after removing coating?

We should point out that the particles in Figure 7 include pure BC and BC without thick coatings. We cannot conclude that smaller particles have more coatings. Instead, we observed that most aged BC particles were small after long-range transport, regardless they were initially small or became smaller due to the collapse of large BC aggregates. We did not quantify the size change after removing coating due to the limitation of the employed TEM instrument. We have now revised the main text in lines 364-374, "The BC particles showed narrower Feret diameters (229–2557 nm) during navigation than those (78-2926 nm) of BC from the own ship during stop. The $D_f$ values during navigation were in a range of 1.28–1.77 with a median of 1.61, while the $D_f$ values during stop were 1.43–1.76 with a median of 1.61, indicating no significant differences of $D_f$ for the exposed BC particles during navigation and stop. Note that the particles in Figure 7 include pure BC and BC without thick coatings. These particles were exposed to the electron beam and volatile coatings were removed so that the morphology of BC was clearly shown regardless of the mixing state of the original BC particles (Figure S7). Most BC particles were below 1 µm in Feret diameter during navigation (Figure

7), while their sizes cover a wide range below 3 µm during stop, implying that the aged BC particles become smaller after long-range transport. Despite only a total of 134 BC data points shown in Figure 7, the results are still statistically meaningful due to the wide range of BC sizes covered in our analysis. Note that the size change of a BC particle cannot be determined because the original size of the particle is unknown before the removal of the coatings."

19.    L296-298, "Comparatively, … particles." I did not see significantly difference in lacunarity by looking at the figure. I suggest making a plot as size change vs lacunarity to support your statement.

We have addressed this point in the question above. Since size change could not be determined, we cannot provide a plot of size change vs lacunarity as suggested by the reviewer.

20.    Figure 7, I cannot see your tar ball. Please mark them in your figures. Also, the scales and text in figures a-c are very difficult to read. Please change a color. Same comments for Figure S9. Fig. 7c looks like thick OC coated soot since I did not see any beam damage, which is typically generated during engine emission. Do you refer amorphous carbon agglomerates to OC or soot?

Per the reviewer's suggestion, we have revised the images in Figures 8 and S8, along with their captions. In addition, we have now added a sentence in lines 380-381 in the revision, "In contrast, the particles collected on June 01 were found to be amorphous carbon agglomerates (Fig. 8c and f) which were referred to OC."

21.    3-3.4, "The difference … origin." Which difference you are referring here? Size, number, shape, or something else?

We have now clarified this point in lines 383-384 in the revision, "The shape difference between the tar ball spheres and the amorphous carbon agglomerates may be related to the type of biomass burning or the origin of the ship engines."

22.    Section 3.3. I feel it might be better to move Section 3.3 before Section 3.1.

We agree with the reviewer and Section 3.3 have been merged into Section 3.1. We have also rearranged all the figures accordingly.

23.    L327-347, "The BC concentrations … Sun et al., 2023)." This paragraph does not fit here and should be moved to section 3.5. BC from AE33 does not agree with OC/EC, but their trend agrees. Moreover, I am not sure how could you get optical EC time resolution of 1 min since that should be only measured before thermal process. The R square is also very low for the fitting of AE33 BC and optical EC. Higher AE33 BC and optical EC is because overestimation by assuming only BC absorbing at long wavelength and multi-scattering effects.

We agree with the reviewer and have now moved this paragraph. We have addressed the concern regarding optical EC in the main question #1 (3). Furthermore, the Sunset OC/EC analyzer determines optical EC by continuously monitoring laser transmission data at a wavelength of 660 nm through a quartz filter over the analysis duration. The optical EC data are automatically saved with a time interval of 1 minute by the instrument internal software. More detail on the optical EC measurement can be found in Bauer et al. (2012).

24.    Figure 9 is not clear to me. What is the x axis? Should you also have a box plot for EC rather than a single value? You can show two plots (one for OC/EC ratio and the other one for EC) for all periods combined. The whisker should not touch axis. Also, I suggest using violin plot instead of box so that you can show distribution.

We thank the reviewer for the suggestion and we have now modified Figure 9 using violin plots to show the median and distribution of both OC/EC ratios and EC concentrations in lines 395-396, "Figure 9 shows the distribution of the OC/EC ratios and the corresponding EC concentrations" and in lines 399-402, "Compared with Figure 9d, the scattered higher OC/EC ratios in Figure 9a/b/c are caused by the very low EC concentrations. The presence of extremely low EC concentrations, often falling below or near the detection limit, can introduce discrepancies in the calculation of the OC/EC split, ultimately resulting in inaccurate EC concentrations (Bauer et al., 2009)."

25.    L333-334, "Notably … during SPP." Please add uncertainties.

Here, we show the median rather than the mean of the mass concentrations so we don't think we can provide uncertainties for this statement.

Figure 10. Please add more tick labels in b and c since current version does not tell the timestamp.

We have updated Figure 10(b, c) according to the reviewer's suggestion.

26.    369-371, "Notably, … (Fig. 10a)." Why do you have a range of BC mass concentration? Is this the BC mass concentration at each wavelength? AE33 reports mass equivalent to the mass of BC absorbs same amount of light, not real BC mass.

We agree with the reviewer that the reported mass is the mass equivalent to the BC mass absorbed at certain wavelengths. We have now modified the text in lines 435-440 in the revision and emphasized those concentrations are wavelength- dependent, "The BC mass concentration ranged from 1.45 to 3.62 μg m$^{-3}$ during biomass burning events based on light absorption at wavelength of 880 nm. The mass concentration in Figure 10 corresponds to BC mass concentration obtained at each wavelength. We have emphasized that BC mass concentration in this study is equivalent BC at individual wavelength. Notably, efficient light absorption of BrC in the range at 370–660 nm was observed during the biomass burning events, while no significant wavelength-dependent BC concentrations were found during the own ship pollution (Fig. 10a)."

**References (from reviewer):**

Cheng, Z., Atwi, K., Onyima, T., Saleh, R., 2019. Investigating the dependence of light-absorption properties of combustion carbonaceous aerosols on combustion conditions. Aerosol Sci. Technol. 53, 419–434. https://doi.org/10.1080/02786826.2019.1566593

Cheng, Z., Sharma, N., Tseng, K.P., Kovarik, L., China, S., 2021. Direct observation and assessment of phase states of ambient and lab-generated sub-micron particles upon humidification. RSC Adv. 11, 15264–15272. https://doi.org/10.1039/d1ra02530a

Corbin, J.C., Czech, H., Massabò, D., de Mongeot, F.B., Jakobi, G., Liu, F., Lobo, P., Mennucci, C., Mensah, A.A., Orasche, J., Pieber, S.M., Prévôt, A.S.H., Stengel, B., Tay, L.-L., Zanatta, M., Zimmermann, R., El Haddad, I., Gysel, M., 2019.

Infrared-absorbing carbonaceous tar can dominate light absorption by marine-engine exhaust. npj Clim. Atmos. Sci. 2. https://doi.org/10.1038/s41612-019-0069-5

Girotto, G., Bhandari, J., Gorkowski, K., Scarnato, B. V, Capek, T., Marinoni, A., Veghte, D.P., Kulkarni, G., Aiken, A.C., Dubey, M., Mazzoleni, C., 2018. Fractal-like Tar Ball Aggregates from Wildfire Smoke. Environ. Sci. Technol. Lett. 5, 360−365. https://doi.org/10.1021/acs.estlett.8b00229

Lack, D.A., Cappa, C.D., 2010. Impact of brown and clear carbon on light absorption enhancement, single scatter albedo and absorption wavelength dependence of black carbon. Atmos. Chem. Phys. 10, 4207–4220. https://doi.org/10.5194/acp-10-4207-2010

**References (from authors)**

Adachi, K., Sedlacek, A. J., Kleinman, L., Chand, D., Hubbe, J. M., and Buseck, P. R.: Volume changes upon heating of aerosol particles from biomass burning using transmission electron microscopy, Aerosol Sci. Technol., 52, 46-56, 10.1080/02786826.2017.1373181, 2017.

Bauer, J. J., Yu, X.-Y., Cary, R., Laulainen, N., and Berkowitz, C.: Characterization of the Sunset semi-continuous carbon aerosol analyzer, J. Air Waste Manag. Assoc., 59, 826-833, 10.3155/1047-3289.59.7.826, 2009.

Brown, S., Minor, H., O'Brien, T., Hameed, Y., Feenstra, B., Kuebler, D., Wetherell, W., Day, R., Tun, R., Landis, E., and Rice, J.: Review of Sunset OC/EC instrument measurements during the EPA's Sunset carbon evaluation project, Atmosphere (Basel), 10, 287, 10.3390/atmos10050287, 2019.

Chen, X., Ye, C., Wang, Y., Wu, Z., Zhu, T., Zhang, F., Ding, X., Shi, Z., Zheng, Z., and Li, W.: Quantifying evolution of soot mixing state from transboundary transport of biomass burning emissions, iScience, 26, 108125, 10.1016/j.isci.2023.108125, 2023.

Dong, Z., Kang, S., Qin, D., Shao, Y., Ulbrich, S., and Qin, X.: Variability in individual particle structure and mixing states between the glacier–snowpack and atmosphere in the northeastern Tibetan Plateau, The Cryosphere, 12, 3877-3890, 10.5194/tc-12-3877-2018, 2018.

Drinovec, L., Močnik, G., Zotter, P., Prévôt, A. S. H., Ruckstuhl, C., Coz, E., Rupakheti, M., Sciare, J., Müller, T., Wiedensohler, A., and Hansen, A. D. A.: The "dual-spot" Aethalometer: An improved measurement of aerosol black carbon with real-time loading compensation, Atmos. Meas. Tech., 8, 1965-1979, 10.5194/amt-8-1965-2015, 2015.

Ezani, E., Dhandapani, S., Heal, M. R., Praveena, S. M., Khan, M. F., and Ramly, Z. T. A.: Characteristics and source apportionment of black carbon (BC) in a suburban area of Klang Valley, Malaysia, Atmosphere, 12, 10.3390/atmos12060784, 2021.

Laing, J. R., Jaffe, D. A., and Sedlacek, I. I. I. A. J.: Comparison of Filter-based Absorption Measurements of Biomass Burning Aerosol and Background Aerosol at the Mt. Bachelor Observatory, Aerosol Air Qual. Res., 20, 663-678, 10.4209/aaqr.2019.06.0298, 2020.

Liu, X., Zhu, R., Jin, B., Zu, L., Wang, Y., Wei, Y., and Zhang, R.: Emission characteristics and light absorption apportionment of carbonaceous aerosols: A tunnel test conducted in an urban with fully enclosed use of E10 petrol, Environ. Res., 216, 10.1016/j.envres.2022.114701, 2023.

Song, X., Xie, X., Qiu, B., Cao, H., Xie, S.-P., Chen, Z., and Yu, W.: Air-Sea Latent Heat Flux Anomalies Induced by Oceanic Submesoscale Processes: An Observational Case Study, Front. Mar. Sci., 9, 10.3389/fmars.2022.850207, 2022.

Yus-Díez, J., Bernardoni, V., Močnik, G., Alastuey, A., Ciniglia, D., Ivančič, M., Querol, X., Perez, N., Reche, C., Rigler, M., Vecchi, R., Valentini, S., and Pandolfi, M.: Determination of the multiple-scattering correction factor and its cross-sensitivity to scattering and wavelength dependence for different AE33 Aethalometer filter tapes: a multi-instrumental approach, Atmos. Meas. Tech., 14, 6335-6355, 10.5194/amt-14-6335-2021, 2021.

Zhao, G., Yu, Y., Tian, P., Li, J., Guo, S., and Zhao, C.: Evaluation and Correction of the Ambient Particle Spectral Light Absorption Measured Using a Filter-based

Aethalometer, Aerosol Air Qual. Res., 20, 1833-1841, 10.4209/aaqr.2019.10.0500,

2020.